# Glycerol 3-phosphate phosphatase/PGPH-2 counters metabolic stress and promotes healthy aging via a glycogen sensing-AMPK-HLH-30-autophagy axis in *C. elegans*

Elite Possik[1,6] ✉, Laura-Lee Klein[1], Perla Sanjab[1], Ruyuan Zhu[1,2], Laurence Côté[1], Ying Bai [1,2], Dongwei Zhang [3], Howard Sun [1], Anfal Al-Mass [1,3], Abel Oppong[1], Rasheed Ahmad [4], Alex Parker [5], S.R. Murthy Madiraju [1], Fahd Al-Mulla [4] & Marc Prentki [1] ✉

Metabolic stress caused by excess nutrients accelerates aging. We recently demonstrated that the newly discovered enzyme glycerol-3-phosphate phosphatase (G3PP; gene *Pgp*), which operates an evolutionarily conserved glycerol shunt that hydrolyzes glucose-derived glycerol-3-phosphate to glycerol, counters metabolic stress and promotes healthy aging in *C. elegans*. However, the mechanism whereby G3PP activation extends healthspan and lifespan, particularly under glucotoxicity, remained unknown. Here, we show that the overexpression of the *C. elegans* G3PP homolog, PGPH-2, decreases fat levels and mimics, in part, the beneficial effects of calorie restriction, particularly in glucotoxicity conditions, without reducing food intake. PGPH-2 overexpression depletes glycogen stores activating AMP-activate protein kinase, which leads to the HLH-30 nuclear translocation and activation of autophagy, promoting healthy aging. Transcriptomics reveal an HLH-30-dependent longevity and catabolic gene expression signature with PGPH-2 overexpression. Thus, G3PP overexpression activates three key longevity factors, AMPK, the TFEB homolog HLH-30, and autophagy, and may be an attractive target for age-related metabolic disorders linked to excess nutrients.

Glucotoxicity caused by excess glucose levels contributes to the development of aging-related pathologies such as obesity, diabetes and cardiovascular disease[1]. High glucose levels shorten the lifespan in multiple organisms, including *Caenorhabditis elegans (C. elegans)*[2–4]. At the molecular level, excessive amounts of glucose heighten the levels of glycerol-3-phosphate (Gro3P), a key metabolite at the intersection of carbohydrate and lipid metabolism. Gro3P regulates the flux of various metabolic pathways, particularly, the glycerolipid/free fatty acid cycle associated with metabolic and aging disorders[1,5]. Gro3P constitutes the backbone to form triglycerides with three free fatty acyl-CoA[1,5]. The accumulation of Gro3P, following excess nutrients, leads to ectopic fat accumulation. It can also cause metabolic stress,

[1]Departments of Nutrition, Biochemistry and Molecular Medicine, Université de Montréal, Montreal Diabetes Research Center, CRCHUM, Montreal, Canada. [2]Diabetes Research Center, Beijing University of Chinese Medicine, 100029 Beijing, China. [3]Department of Biological Sciences, Faculty of Science, Kuwait University, 13060 Kuwait City, Kuwait. [4]Departments of Immunology, Microbiology, Genetics, and Bioinformatics, Dasman Diabetes Institute, Kuwait City 15462, Kuwait. [5]Department of Neurosciences, CRCHUM, Montreal, Canada. [6]Present address: Department of Medicine, Divisions of Cardiology and Experimental Medicine, McGill University Health Centre (MUHC), Montreal, Canada. ✉e-mail: elite.possik@rimuhc.ca; marc.prentki@umontreal.ca

increase the production of reactive oxygen species (ROS) to damage macromolecules, and cause low-grade chronic inflammation, cellular dysfunction and possibly organismal aging[1,5].

We have recently identified a key evolutionarily conserved enzyme at the heart of metabolism, glycerol-3-phosphate phosphatase (G3PP, gene name *Pgp*), that operates a glycerol shunt by hydrolyzing glucose-derived Gro3P to glycerol[6–9]. In biochemistry textbooks, lipolysis is claimed to be the only source of glycerol. The newly discovered 'glycerol shunt', an alternative pathway to produce glycerol, revamps our understanding of the metabolic basis of gluco-detoxification, and its role in metabolic health, disease, and aging. In pancreatic ß-cells and hepatocytes, we have shown that G3PP regulates glycolysis, glucose and fatty acid oxidation, gluconeogenesis, glycerolipid synthesis, the cellular redox state, and energy production[1,5,6]. In mice, using G3PP pancreatic ß-cell and hepatocyte tissue-specific KO models, we demonstrated that G3PP acts as a "glucose excess detoxification valve" and protects from metabolic stress induced by high nutrients[8,9]. In *C. elegans*, we recently identified three homologs of G3PP (PGPH-1,2,3) and have found that their protein products act as G3PP enzymes and are required for glycerol synthesis and protection from various stresses[7]. Importantly, we have shown that the activation of PGPH-2, the major G3PP worm homolog, mimics some of the beneficial effects of calorie restriction (CR), particularly under glucotoxic conditions, without its associated caveats, such as reduced fertility and food intake. In addition, it extends a healthy lifespan, particularly under glucotoxicity[7]. However, the mechanism whereby the activation of the glycerol shunt via overexpression of PGPH-2/G3PP promotes healthy aging and protects from glucotoxicity in *C. elegans* remains to be determined.

Calorie restriction, the reduction of energy intake below requirements without malnutrition, is the most robust evolutionarily conserved intervention that counters metabolic stress and promotes healthy aging[10–13]. In rodents and non-human primates, CR extends medium and maximum lifespan by up to 50%[10,14–19]. While currently there is no clear evidence of increased longevity by CR in humans, recent clinical trials revealed a rewiring of immunometabolic functions for a longer healthy life[20–22]. Interestingly, a recent exome-wide association study in centenarians has identified PGP/G3PP as a new candidate longevity gene[23].

In the present study, using *C. elegans*, we identify the mechanism whereby the overexpression of PGPH-2 extends a healthy lifespan, particularly under conditions of glucotoxicity. We demonstrate that the activation of the glycerol shunt, via PGPH-2 overexpression, decreases glycogen stores to activate AMP-activated protein kinase (AMPK), which leads to the nuclear translocation of HLH-30 (a *C. elegans* homolog of TFEB) and the induction of autophagy, to promote healthy aging and longevity, particularly under glucotoxic conditions.

## Results

### Stable overexpression of *pgph-2* decreases fat accumulation, extends healthspan and protects from glucotoxicity

In our previous work, we generated transgenic lines overexpressing *pgph-2* and empty vector (EV) via DNA injection with co-injection marker plasmids expressing mCherry in the pharynx to form multi-copy extrachromosomal arrays, which get transmitted to the next generation via partial and variable transmission rates and mosaic expression in different cell types[7,24]. In the current work, we integrated these transgenes into the worm genome to form stable lines overexpressing *pgph-2* or EV used as control and outcrossed them more than seven times to the wild-type (WT) strain and observed that *pgph-2* gene mRNA levels are induced by about six folds (Supplementary Fig. 1a). Similarly to what we have previously observed using extrachromosomal transgene[7], the stable overexpression of *pgph-2* strongly decreased fat accumulation both at normal and excess glucose conditions, without decreasing food intake as measured by the pharyngeal pumping of the animals (Fig. 1a–c). In fact, pharyngeal pumping was increased in *pgph-2 o/e*

animals with age under normal and excess glucose conditions, implying a healthier aging worm (Fig. 1b, c). In *C. elegans*, vitellogenins are large lipo-glyco-phosphoproteins that belong to the yolk family and are primary synthesized in the intestine and transported to oocytes to sustain embryogenesis[25]. During aging, post-reproductive ectopic yolk accumulation largely participates in the loss of worm tissue integrity and function. Longevity interventions such as low insulin signaling, calorie restriction or TOR inhibition were demonstrated to reduce vitellogenin synthesis and accumulation with age[25]. Using the vitellogenin reporter *vit-2*::GFP, we observed that yolk particles are strongly decreased in *pgph-2 o/e* animals (Supplementary Fig. 1b). Glucose feeding strongly increased the accumulation of vitellogenin in ectopic tissues (Supplementary Fig. 1b). However, despite excess glucose feeding, *pgph-2 o/e* animals maintained levels of fat and yolk particles similar to WT animals grown on normal growth conditions, demonstrating that the activation of PGPH-2 protects from nutrient-induced lipid and yolk accumulation (Fig. 1a and Supplementary Fig. 1b).

The accumulation of fat in ectopic tissues has detrimental consequences on the aging of tissues[26]. In line with a decreased ectopic fat accumulation and similarly to what we have previously shown[7], the overexpression of *pgph-2* in animals grown under normal conditions improved the locomotion behavior with age, with a non-significant effect on lifespan (Fig. 1d, e). However, overexpression of *pgph-2* significantly improved healthspan (locomotion) and medium and maximal lifespan under glucotoxic conditions (Fig. 1f, g and Supplementary Data 1). Overall, the results demonstrate that *pgph-2 o/e* mimics some of the key beneficial effects of CR during aging and particularly under glucotoxic conditions (improved locomotion, decreased fat and vitellogenin accumulation, maintenance of food intake) under control conditions, and also increases medium and maximal lifespan under glucotoxicity. In addition, the newly generated *pgph-2* overexpressing stable transgenic line phenocopies are what we have previously observed using the *pgph-2* overexpressing extrachromosomal array transgenes[7].

### PGPH-2 protects from glucotoxicity via mechanisms independent of insulin-like signaling and canonical dietary restriction

Following the characterization of the newly generated transgenic lines, we aimed to identify the mechanism whereby the overexpression of *pgph-2* protects from glucotoxicity and extends healthy lifespan. Calorie restriction (CR) and low insulin signaling are widely known to extend healthspan and lifespan and protect from glucotoxicity across taxa, including *C. elegans*[27]. To determine whether the mechanism of protection from glucotoxicity mediated by *pgph-2 o/e* depends on the canonical CR pathway, we used mutant animals that harbor a mutation in the nicotinic acetylcholine subunit *eat-2* in the pharynx, which causes defective feeding behavior[28]. Unlike what we have previously shown in the extrachromosomal transgene model, stable *pgph-2 o/e* caused a mild but significant decrease in brood size, both under normal and glucose excess conditions, likely due to a stronger, more stable and ubiquitous expression (7 folds vs 4 folds increased mRNA levels) (Supplementary Fig. 1a, c). Noticeably, the overexpression of *pgph-2* led to a smaller decrease in brood size in comparison to the severe brood size decrease observed in *eat-2* animals in normal growth and excess glucose conditions (Supplementary Fig. 1d). Loss of *eat-2* in *pgph-2 o/e* animals strongly decreased brood size and led to very small and transparent animals. The additive effects of *pgph-2 o/e* and *eat-2* on brood size reduction support the concept that the *pgph-2 o/e* pathway causing reduced brood size is independent of classical CR (Fig. 1h). Diminished insulin signaling by mutation of the IGF1 receptor homolog DAF-2 extends lifespan and healthspan via the FOXO homolog, DAF-16[29]. Interestingly, *pgph-2 o/e* extends the healthspan and lifespan of the short-lived *daf-16 (mu86)* mutant, also suggesting that the protection from glucotoxicity mediated by *pgph-2 o/e* is independent of the insulin-like signaling pathway (Fig. 1i, j and Supplementary Data 1).

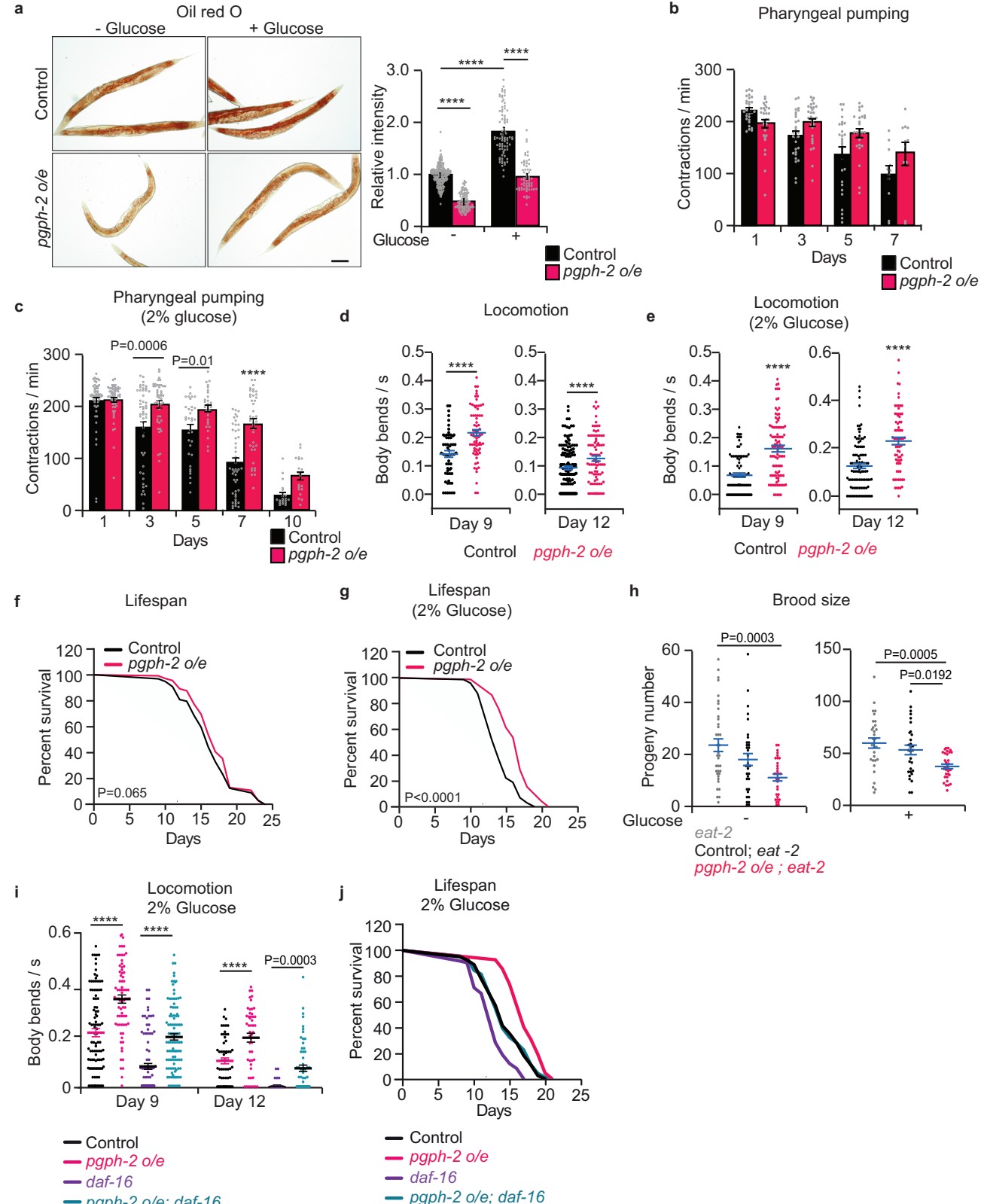

### PGPH-2 o/e promotes healthspan and lifespan under glucotoxic conditions via HLH-30

TFEB is a master transcriptional regulator of lysosomal and autophagy genes, particularly in conditions of CR and starvation, to rewire metabolism and ensure adaptation and survival[30–35]. The worm homolog of TFEB, HLH-30, is constitutively activated in known longevity mutants involving low insulin signaling, mild mitochondrial impairment and CR, and has been shown to promote survival under starvation and infection with various pathogens[31,36–38]. To assess whether *pgph-2* o/e protects from glucotoxicity via HLH-30, we measured the HLH-30 nuclear translocation using the *hlh-30p::hlh-30::GFP* reporter. Importantly, *pgph-2* o/e increased the percentage of young adult animals with constitutive HLH-30 nuclear translocation, both under normal and excess glucose conditions (Fig. 2a). The enhanced

**Fig. 1 | Stable overexpression of *pgph-2* decreases fat accumulation, extends healthspan and protects from glucotoxicity.** For all figure panels, **** represents *P < 0.0001*. **a** Oil red O staining and quantification in Control and *pgph-2* o/e 1-day adult animals treated with or without 2% glucose. Data represent mean ± SEM, n = 3 independent experiments. Scale bar = 50 μm. *P*-values are obtained using one-way ANOVA with the Bonferroni test. **b, c** Bar plots showing pharyngeal pumping rates of control and *pgph-2* overexpressing nematodes at indicated days of age at normal growth conditions (**b**) and 2% glucose conditions (**c**). Data represent mean ± SEM, n = 3 independent experiments. *P*-values are obtained by two-way ANOVA with Bonferroni correction. **d, e** Body bends per second of *pgph-2* o/e animals in comparison to controls in normal growth (**d**) and 2% glucose conditions (**e**). Data is shown using dot plots with denoted mean ± SEM, n = 3 independent experiments. The number of tracks is shown in datasource. *P*-values are obtained by two-tailed unpaired Student's *t*-test. **f, g** Lifespan of *pgph-2* overexpressing animals in comparison to control nematodes in normal growth conditions (**f**) and on plates

supplemented with 2% glucose (**g**). The number of separate lifespan experiments, animals and detailed statistics are shown in Supplementary Data 1. *P*-value is obtained using the two-sided Mantel–Cox test. **h** Brood size as measured by the number of progeny per animal in *eat-2 (ad465)*; control; *eat-2 (ad465)*, and *pgph-2 o/e*; *eat-2 (ad465)* grown on normal growth medium or plates supplemented with 2% glucose. Data are shown as dot plots representing mean ± SEM from three independent experiments. *P*-values are obtained by one-way ANOVA with the Bonferroni test. **i** Locomotion analysis on days 9 and 12 of age, measured by body bends per second in indicated strains grown on plates supplemented with 2% glucose. Data represent mean ± SEM, n = 3 independent experiments. *P*-values are obtained by one-way ANOVA with the Bonferroni test. **j** Lifespan of control, *pgph-2 o/e*, *pgph-2 o/e*; *daf-16 (mu86)*, and *daf-16 (mu86)* overexpressing animals on plates supplemented with 2% glucose. The number of experiments, animals and detailed statistics are shown in Supplementary Data 1. *P*-values are obtained using the two-sided Mantel–Cox test.

nuclear translocation of HLH-30 mediated by *pgph-2 o/e* is not due to a transgene artifact since control mCherry transgenic animals showed low levels of HLH-30 nuclear translocation similar to the reporter HLH-30 animals with a WT background (Supplementary Fig. 1e). To directly assess the role of HLH-30 in the decreased fat accumulation or protection from glucotoxicity phenotypes, we crossed the *pgph-2 o/e* and control strains to *hlh-30* mutant animals. HLH-30 was found to be required for the decreased fat accumulation in *pgph-2* o/e animals since its loss restored fat levels to normal values under normal conditions (Fig. 2b). Under glucose excess conditions, loss of *hlh-30* also restored fat levels but only partially, suggesting that additional pathways contribute to the decreased fat deposition observed in *pgph-2 o/e* animals (Fig. 2c). Importantly, loss of *hlh-30* abrogated the increased healthspan as measured by locomotion at days 9 and 12, and lifespan under glucotoxic conditions in *pgph-2 o/e* animals (Fig. 2d, e and Supplementary Data 1). Loss of *hlh-30* also abolished the *pgph-2 o/e*-mediated increased locomotion under glucose excess conditions at day 4, an earlier time point where the animals with the mutant phenotypes are in a healthier state, ascertaining that HLH-30 is required for the locomotion benefits mediated by *pgph-2 o/e* and that the results obtained on days 9 and 12 of age are not due to the fact that *hlh-30* mutant animals are short-lived and sick at an old age (Supplementary Fig. 3a). Overall, the data demonstrate that HLH-30 is activated by *pgph-2* o/e and is required for the beneficial effects on healthspan and lifespan under glucose excess.

### RNA-seq analysis reveals catabolic and longevity signatures with *pgph-2* o/e

To assess the genes differentially regulated by *pgph-2* and that could provide insight into the understanding of the glycerol shunt in healthy aging, and to determine the subset of *pgph-2*-regulated genes that may be dependent on HLH-30, we performed whole animal RNA-seq from synchronized WT, *pgph-2 o/e*, *hlh-30* and *pgph-2 o/e*; *hlh-30* animals under basal conditions and glucose excess (Fig. 3a). Principal component analysis, gene density distributions and Venn diagrams showing overlapping genes among RNA-seq datasets are indicated in Supplementary Fig. 2. We identified 669 genes differentially expressed in *pgph-2 o/e* in comparison to WT with 349 upregulated and 320 downregulated (Fig. 3b and supplementary Data 2). Gene ontology analysis of the upregulated genes indicates enrichment of genes belonging to catabolic processes, such as 'hydrolase activity', 'lipid catabolic processes', 'lipid metabolism', 'lysosome', and 'lytic vacuoles' (Fig. 3c and Supplementary Data 3). Noticeably, innate immune response terms such as 'defense to bacterium' or 'defense to external stimuli' are also enriched, supporting a role for *pgph-2* in the promotion of healthy aging[39–42] and the possible involvement of *hlh-30* in this response (Fig. 3c). Thus, HLH-30 is known to play a key role in the induction of innate immunity genes, particularly in cases of pathogen infection[36–38,43]. Moreover, starvation and calorie

restriction interventions are known to activate the innate immune response[38,42]. Under glucose excess conditions, 3263 genes were differentially expressed compared to WT, illustrating a major phenotypic change under glucotoxicity (Supplementary Fig. 3b and Supplementary Data 4). Gene ontology analysis of genes upregulated in *pgph-2 o/e* animals in comparison to WT under 2% excess glucose conditions also revealed similar changes as without glucose such as 'innate immune response', 'carbohydrate binding', 'unfolded protein response', and 'stress response' genes, suggesting that while the longevity outcomes are different, *pgph-2 o/e* is exerting similar gene expression changes under normal and glucose excess conditions (Supplementary Fig. 3c, d and Supplementary Data 5). We also identified 60 genes upregulated in *pgph-2 o/e* animals simultaneously in normal and glucose excess conditions belonging to 'lipid lipases', 'catabolic processes' and 'innate immune response categories,' further suggesting that the healthy longevity mechanisms under basal and excess glucose conditions share similarities (Supplementary Fig. 3e and Supplementary Data 6). Of particular interest, gene expression levels of *eat-2* were significantly increased in *pgph-2 o/e* animals in glucose excess conditions impinging on the physiological role of the glycerol shunt as a CR mimetic, in particular under glucotoxic conditions (Supplementary Fig. 3f). Importantly, the comparison of the list of genes upregulated in *pgph-2 o/e* animals to previously published lists of genes upregulated in the longevity mutants as *daf-2* and *eat-2* highlights 48 common genes belonging to the innate immune response, catabolic activity, and longevity-related mechanisms (Supplementary Fig. 3g and Supplementary Data 7)[44]. Overall, the RNA-seq data support the view that *pgph-2 o/e* governs a healthy aging gene expression signature that overlaps significantly with the signatures of other longevity mutants.

### HLH-30 is a key regulator of the gene expression signature mediated by *pgph-2* o/e

We observed that 214 of the 349 genes upregulated in *pgph-2 o/e* vs WT were downregulated in the double mutant *pgph-2 o/e*; *hlh-30* vs *pgph-2 o/e*, and the heat map analysis of differentially expressed genes under normal conditions revealed large clusters of genes enriched in *pgph-2 o/e* animals in an *hlh-30*-dependent manner, suggesting that HLH-30 plays a key role in the gene expression signature mediated by the overexpression of *pgph-2* (Fig. 3d, e and Supplementary Data 8). Of particular interest, several of these genes belong to the lysozyme families, including *ilys-2*, *lys-4*, *lys-5*, *lys-6*, *lys-7*, suggesting enhanced lysosomal hydrolase activity. Moreover, several lipid metabolism, innate immune response, carbohydrate binding and hydrolase genes were increased in *pgph-2 o/e* animals in an *hlh-30*-dependent manner (Fig. 3e and Supplementary Data 9). In addition to its role as a master transcriptional regulator of lysosomal biogenesis and metabolism genes, HLH-30 regulates the transcription of genes involved in the autophagic process, a self-cleaning and recycling process, normally

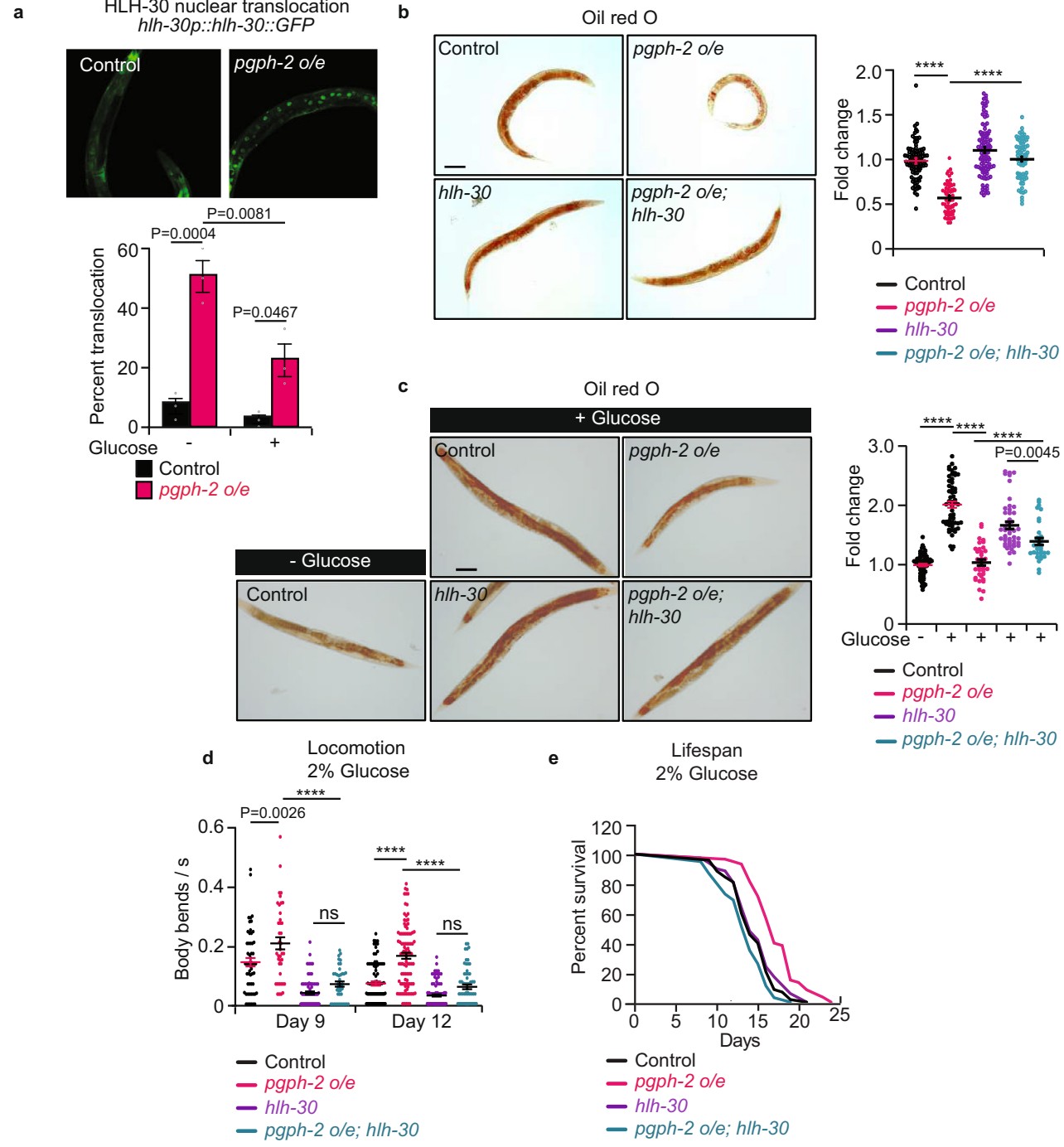

**Fig. 2 | PGPH-2 o/e activates the nuclear translocation of HLH-30 and promotes healthy aging under glucotoxic conditions.** For all panels, **** represents *P < 0.0001*. **a** Percent nuclear translocation of HLH-30 in young adult animals scored using the HLH-30::GFP expressing strain crossed to control and *pgph-2 o/e* animals under normal growth conditions or 2% glucose. Data represent mean ± SEM from three independent experiments. *P*-values are obtained by one-way ANOVA with the Bonferroni correction. **b**, **c** Oil red O staining and quantification in control, *pgph-2 o/e, pgph-2 o/e; hlh-30 (tm1978)*, and *hlh-30 (tm1978)* strains on normal growth conditions at day 1 (**b**) and 2% glucose conditions at day 3 (**c**). Data is shown using dot plots with denoted mean ± SEM from three independent experiments. *P*-

values are obtained by one-way ANOVA with the Bonferroni correction. The scale bar = 50 μm. **d** Locomotion analysis on days 9 and 12 of age, measured by body bends per second control, *pgph-2 o/e, pgph-2 o/e; hlh-30 (tm1978)*, and *hlh-30 (tm1978)* strains grown on plates supplemented with 2% glucose. Data represent mean ± SEM from three independent experiments. *P*-values are obtained by one-way ANOVA with the Bonferroni correction. The number of tracks is shown in datasource. **e** Lifespan of control, *pgph-2 o/e, pgph-2 o/e; hlh-30 (tm1978)*, and *hlh-30 (tm1978)* overexpressing animals on plates supplemented with 2% glucose. The number of separate lifespan experiments, animals and detailed statistics are shown in Supplementary Data 1. *P*-value is obtained using the two-sided Mantel−Cox test.

activated in low nutrient conditions such as calorie restriction and starvation to produce energy and promote survival[31,34,45–49]. The *atg-4.1, lgg-2* (LC3 homolog) and *atg-18 genes*, which are key components of the autophagy machinery, were dependent on HLH-30 for their

upregulation caused by *pgph-2 o/e* (Supplementary Fig. 4a and Supplementary Data 10). Overall, the data support the view that many genes induced by *pgph-2* overexpression require HLH-30 for their induction.

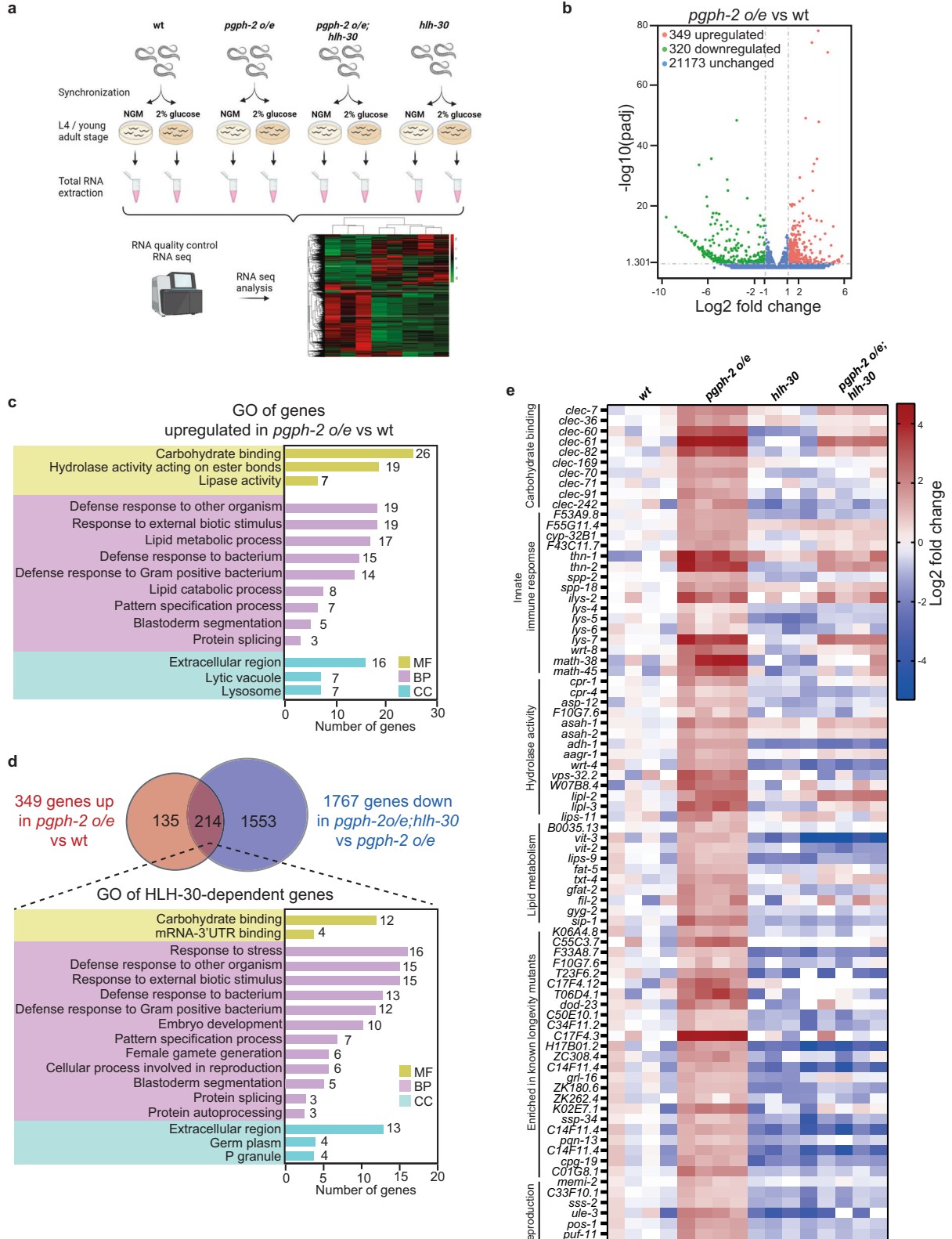

**a**

**b** *pgph-2 o/e* vs wt

**c** GO of genes upregulated in *pgph-2 o/e* vs wt

**d** 349 genes up in *pgph-2 o/e* vs wt — 135 | 214 | 1553 — 1767 genes down in *pgph-2o/e;hlh-30* vs *pgph-2 o/e*

GO of HLH-30-dependent genes

**e**

## PGPH-2 o/e activates autophagy in a TFEB/HLH-30-dependent manner

Recent extensive research highlights a critical role for autophagy in the promotion of healthy aging and the protection from metabolic and aging diseases[31,34,45–49]. Because HLH-30 appears to exert a critical role in the rewiring of gene expression in *pgph-2* o/e animals, we hypothesized that *pgph2 o/e* promotes healthy aging and protects from

glucotoxicity via HLH30-dependent autophagy activation. Using the intestinal *LGG-1::mCherry* autophagy reporter, we observed that *pgph-2 o/e* increases autophagy both under normal and glucose-induced conditions (Fig. 4a, b). The increased autophagy in *pgph-2* o/e transgenic animals was not due to the transgene itself as control mCherry animals showed autophagy levels equivalent to WT (Supplementary Fig. 4b). To determine whether the accumulation of autophagosomes

**Fig. 3 | RNA-seq analysis reveals catabolic and longevity signatures specific to *pgph-2 o/e* and largely dependent on HLH-30. a** Schematic representation of the RNA-seq experimental design. Briefly, synchronized WT, *pgph-2 o/e*, *pgph-2 o/e; hlh-30* and *hlh-30* animals and grown on normal growth medium or plates supplemented with 2% glucose are harvested. Total RNA is extracted using trizol and purified and subjected to seq analysis. Scheme created with Biorender.com. **b** Volcano plots showing differentially expressed genes in WT vs *pgph-2 o/e* animals on normal growth medium. Genes with a fold change >2 and a *P*-value smaller than 0.05 were considered significantly changed. Red, green, and blue indicate genes that are significantly upregulated, significantly downregulated, or not significantly changed, respectively. **c** Gene ontology (GO) annotations of genes significantly upregulated in *pgph-2 o/e* animals in comparison to WT and belonging to the molecular function (MF), biological processes (BP), and cellular component (CC) categories. **d** Venn diagram and gene ontology annotations of the common genes that are upregulated in *pgph-2 o/e* animals in comparison to WT and downregulated in *pgph-2 o/e; hlh-30* in comparison to *pgph-2 o/e*. **e** Heat map representing the differential expression of selected genes based on functions related to longevity in WT, *pgph-2 o/e*, *pgph-2 o/e; hlh-30* and *hlh-30* animals. Carbohydrate binding, innate immune response, hydrolase activity, lipid metabolism, longevity and reproduction are upregulated in *pgph-2 o/e* animals in an HLH-30-dependent manner.

promoted by *pgph2 o/e* is due to active autophagic flux rather than impaired flux, we employed the dual tandem-tagged (mCherry/GFP) form of LGG-1 that enables the distinction between autophagosomes and autolysosomes[50] (Supplementary Fig. 5a). Because the GFP tag is acid-sensitive and the mCherry tag is acid-insensitive, following the fusion of the autophagosome with the lysosome that leads to active autolysosomes, GFP is quenched, and only the mCherry signal is emitted. Therefore, in active autolysosomes red-only fluorescence is emitted, while in autophagososomes both signals are emitted (Supplementary Fig. 5a). We found a high number of autolysosomes (red puncta) in *pgph-2 o/e* animals in comparison to controls (Supplementary Fig. 5b). Importantly, blocking autophagy using *atg-7* RNAi, which removes a key component of the autophagy machinery impaired the autophagic flux to a similar extent both in *control* and *pgph-2 o/e* animals and led to a strong accumulation of autophagosomes (yellow puncta), indicating that the autophagic flux in *pgph-2 o/e* animals is more active and can be impaired by *atg-7* suppression (Supplementary Fig. 5b). Notably, in *pgph-2 o/e* animals autophagy was also induced in tissues other than the intestine, including muscle, vulval structure and hypodermal seam cells (Supplementary Fig. 6a). To determine whether HLH-30 is required for the enhanced autophagic activity mediated by *pgph-2 o/e*, we crossed the WT and *pgph-2 o/e* autophagy reporter strains to *hlh-30* mutants and assessed the number of autophagosomes in intestinal cells (Fig. 4c). We found that HLH-30 is required for the increased autophagic activity mediated by *pgph-2 o/e* both under normal and glucose excess conditions since the loss of *hlh-30* reduced the enhanced autophagic activity of *pgph-2 o/e* animal to a level comparable to single *hlh-30* mutant animals (Fig. 4c). Moreover, loss of *atg-18*, an important protein in the autophagic vesicle elongation process[46], suppressed the healthspan and lifespan extension promoted by *pgph-2 o/e* under glucose excess conditions (Fig. 4d, e, Supplementary Fig. 4c and Supplementary Data 1). Loss of *atg-18* also abolished the *pgph-2 o/e*-mediated increased locomotion under glucose excess conditions also at day 4, an earlier time point where the animals with the mutant phenotypes are in a healthier state (Fig. 4d and Supplementary Fig. 4c). Altogether, the results demonstrate that *pgph-2* overexpression increases autophagy in an *hlh-30*-dependent manner which is required for the *pgph-2o/e*-mediated extension of healthspan and lifespan under glucotoxic conditions.

## PGPH-2 o/e activates the AMPK-HLH-30-autophagy signaling cascade to promote healthy aging under glucotoxic conditions

Several longevity-promoting pathways and notably the TOR nutrient-sensing pathway are primary regulators of TFEB/HLH-30 activity[31,51]. Under anabolic conditions, TOR phosphorylates TFEB and sequesters it in the cytoplasm, preventing the transcription of the CLEAR (Coordinated Lysosomal Expression and Regulation) network. In contrast, under catabolic conditions such as calorie restriction, TOR is inhibited, blocking energy-consuming pathways and activating energy-generating pathways to promote survival[32,34]. Under catabolic conditions, a $Ca^{2+}$ flux to the cytosol derived from lysosomes activates calcineurin and leads to the dephosphorylation of TFEB and its translocation to the nucleus[32,34]. In the worm, TOR inhibition activates HLH-30 and extends healthspan and lifespan[31,52]. Using the *hlh-30p::hlh-30::GFP* reporter strain, we recapitulated that TOR inhibition increases nuclear translocation of HLH-30[31,43,51]. However, our data demonstrates that in animals with TOR suppression by RNAi, *pgph-2 o/e* further increases the HLH-30 nuclear translocation, suggesting that *pgph-2* o/e activates HLH-30 in a TOR-independent manner (Fig. 5a). Moreover, brood size was further decreased in *pgph-2 o/e* animals treated with TOR RNAi in comparison to *pgph-2 o/e* animals treated with empty vector (EV) RNAi, further supporting that the longevity pathway mediated by *pgph-2 o/e* is independent of TOR inhibition (Fig. 5b).

TFEB activation pathways other than the initially discovered TOR inhibition pathway have recently been discovered. Particularly, AMPK has been shown to phosphorylate TFEB in mammalian cells driving its nuclear translocation[53]. Our data indicate an increased activation of AMPK as phospho-thr172 levels are increased by overexpression of PGPH-2 (Fig. 5 and supplementary Fig. 6b). In *C. elegans*, loss of the AMPK catalytic subunit *aak-2*, suppresses the translocation of HLH-30 during *S. aureus* infection[43]. We observed that loss of *aak-2* suppresses the increased HLH-30 nuclear translocation mediated by *pgph-2 o/e* both under normal and glucose excess conditions (Fig. 5d). Moreover, loss of *aak-2* completely suppressed the increased autophagic activity mediated by *pgph-2* o/e both under normal and excess glucose conditions (Fig. 5e, f). Importantly, AAK-2 is required for the protection from glucotoxicity mediated by *pgph-2 o/e* since its loss, strongly suppressed the extension of lifespan and healthspan phenotypes under glucotoxic conditions (Fig. 5g, h and Supplementary Data 1). Overall, the data demonstrate that *pgph-2 o/e* activates an AMPK signaling cascade which then transduces a catabolic signal that constitutively induces the HLH-30 nuclear translocation and autophagy to promote healthspan and lifespan under excess glucose condition.

## Metabolomics analysis in *pgph-2* overexpressing animals

To determine whether the activation of the glycerol shunt leads to a severe drop in the levels of amino acids and TCA cycle intermediates in an analogy of a severe effect of calorie restriction, we used a metabolomics approach and assessed metabolite levels in control and *pgph-2 o/e* animals both under normal growth and excess glucose conditions. Since *pgph-2 o/e* activates AMPK and because AMPK is activated by a rise in the AMP/ATP and ADP/ATP ratios[54–56], we aimed to examine whether changes in the AMP, ADP and ATP levels could explain the enhanced activity of the enzyme. However, the AMP, ADP and ATP levels remained unchanged in the absence or presence of excess glucose (Fig. 6), indicating that changes in adenine nucleotide levels cannot explain the enhanced AMPK activity in *pgph-2 o/e* animals. As expected, Gro3P content was reduced upon *pgph-2* overexpression. Examination of other metabolites indicated that the levels of Krebs cycle intermediates, some amino acids, cAMP, NAD, NADH, NADP, NADPH and acetyl-CoA remained unchanged, indicating no gross alteration in intermediate metabolism in these animals (Fig. 6 and Supplementary Fig. 7a). The DHAP/Gro3P and pyruvate/lactate ratios were higher supporting the view that the cytosolic $NAD^+$/NADH ratios were increased to support high glycolytic flux. For reasons which are uncertain, GTP and GDP levels under glucose excess conditions were higher in *pgph-2* o/e

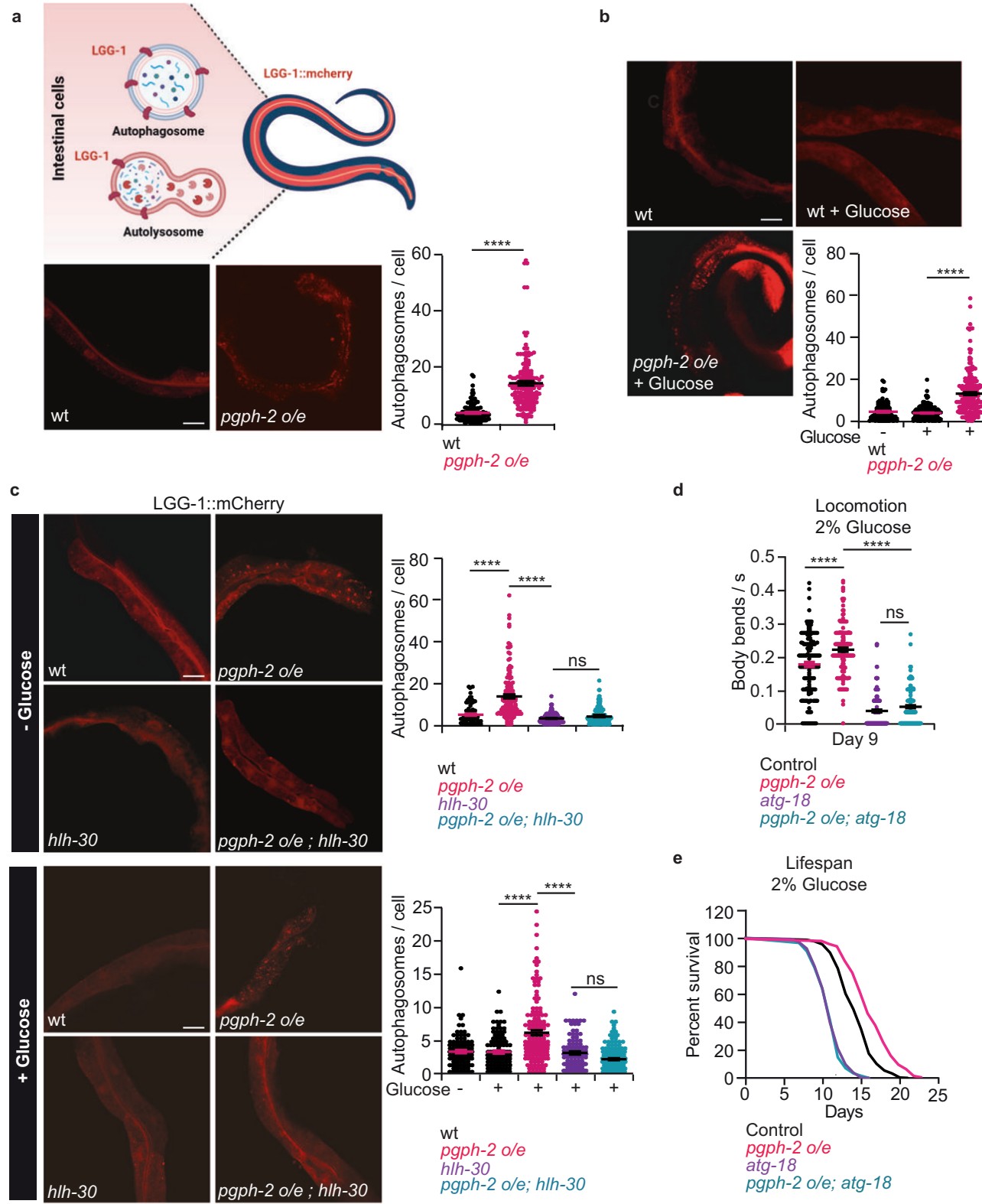

animals, but this also supports the view of enhanced glucose metabolism in these animals. The GSH/GSSG ratio, an antioxidant marker, was significantly increased in *pgph-2 o/e* animals in comparison to the controls under glucotoxic conditions, further supporting enhanced glucose metabolism in these animals under nutrient-rich conditions. Overall, the metabolomics data indicates that *pgph-2 o/e* does not cause reduced intermediate metabolism but rather enhanced metabolism despite the constant drainage of Gro3P to glycerol.

## PGPH-2o/e reduces glycogen stores to activate an AMPK-dependent/HLH-30/autophagy cascade

AMPK is an energy sensor that activates catabolic pathways upon decreases in energy levels[56]. Since the [AMP:ATP] as well as [ADP:ATP] ratios are not increased in *pgph-2 o/e* animals in comparison to controls both under normal and high glucose conditions, we asked how is AMPK activated independently of a change in energy drop? Recent work has shown that the AMPK-mediated energy-sensing mechanism

**Fig. 4 | PGPH-2 o/e activates autophagy in an HLH-30-dependent manner.** For all figure panels, **** represents *P < 0.0001*. **a** Representative confocal images and quantification of LGG-1::mCherry puncta in the intestines of day 1 WT and *pgph-2 o/e* animals under normal growth conditions. The scale bar = 20 μm. Data is shown using dot plots with denoted mean ± SEM from three independent experiments. *P*-values are obtained by unpaired two-tailed Student's *t*-test. Graphical representation created using Biorender.com. **b** Representative confocal images and quantification of LGG-1::mCherry puncta in the intestines of day 1 WT and *pgph-2 o/e* animals grown on plates supplemented with 2% glucose. Data is shown using dot plots with denoted mean ± SEM from three independent experiments. *P*-values are obtained by one-way ANOVA with the Bonferroni correction. The scale bar = 20 μm. **c** Representative confocal images and quantification of LGG-1::mCherry puncta in the intestines of day 1 WT, *pgph-2 o/e*, *pgph-2 o/e; hlh-30 (tm1978)*, and *hlh-30*

*(tm1978)* animals grown on normal growth condition or plates supplemented with 2% glucose. The scale bar = 20 μm. Data is shown using dot plots with denoted mean ± SEM from three independent experiments. *P*-values are obtained by one-way ANOVA with the Bonferroni test. **d** Locomotion analysis on day 9 of age, measured by body bends per second control, *pgph-2 o/e*, *pgph-2 o/e; atg-18 (gk378)*, and *atg-18 (gk378)* strains grown on plates supplemented with 2% glucose. Data is shown using dot plots with denoted mean ± SEM from three independent experiments. *P*-values are obtained by one-way ANOVA with the Bonferroni correction. Number of tracks is shown in datasource. **e** Lifespan of control, *pgph-2 o/e*, *pgph-2 o/e; atg-18 (gk378)*, and *atg-18 (gk378)* overexpressing animals on plates supplemented with 2% glucose. The number of separate lifespan experiments, animals and detailed statistics are shown in Supplementary Data 1. *P*-value is obtained using the two-sided Mantel–Cox test.

is not restricted to a drop in energy levels. Particularly, the regulatory beta subunit of AMPK has a glycogen binding domain[57]. A growing body of research revealed an interchangeable physical and functional interaction between glycogen availability and the AMPK-dependent regulation of whole-body metabolism[57]. Specifically, high glycogen content inhibits AMPK activity, and glycogen depletion activates AMPK[58]. We observed that the overexpression of *pgph-2* strongly decreased glycogen levels both under normal growth and glucotoxic conditions (Fig. 7a, b). To determine whether the catabolic signaling cascade (HLH-30-autophagy) downstream AMPK is activated because glycogen levels are chronically depleted in *pgph-2 o/e* animals, we inhibited glycogen phosphorylase, the enzyme that catalyzes the rate-limiting step of glycogen degradation or glycogenolysis, using *pygl-1* RNAi (Fig. 7c). Inhibition of *pygl-1* abrogated the chronic depletion of glycogen stores in *pgph-2 o/e* animals as glycogen levels were restored to control levels in *pgph-2 o/e* animals treated with *pygl-1* RNAi (Fig. 7d). Also, the inhibition of glycogen degradation significantly decreased the *pgph-2 o/e*-dependent HLH-30 nuclear translocation both under normal and excess glucose conditions (Fig. 7e). Autophagy levels were also strongly reduced in *pgph-2 o/e* animals treated with *pygl-1* RNAi under normal growth and glucotoxic conditions (Fig. 7f). Importantly, inhibition of *pygl-1* abolished the enhanced healthspan both at an early age (day 4; Supplementary Fig. 7b) and older age (Fig. 7g), and lifespan (Fig. 7h) under glucotoxic conditions in *pgph-2 o/e* animals (Supplementary Data 1). Altogether, the results demonstrate that glycogen depletion drives the activation of the downstream AMPK-TFEB-autophagy-healthy aging signaling cascade in *pgph-2 o/e* animals.

## Discussion

Here, we unveil the mechanism by which the activation of PGPH-2/G3PP in *C. elegans* extends healthpsan and lifespan primarily under glucotoxic conditions. We demonstrate that the overexpression of PGPH-2 mimics in part the beneficial effects of CR, particularly under excess glucose, by depleting glycogen stores leading to the constitutive activation of the AMPK-TFEB-autophagy axis, promoting healthy aging and the protection from glucotoxicity (Fig. 8). In brief, the evidence for this is that: (1) PGPH-2 o/e activates AMPK and induces the nuclear translocation of HLH-30. (2) PGPH-2 o/e activates the autophagic flux in an AMPK and HLH-30-dependent manner and loss of AMPK, HLH-30 and autophagy in animals overexpressing *pgph-2* abrogates the enhanced healthy longevity and protection from glucotoxicity phenotypes. (3) The PGPH-2-mediated regulation of the HLH-30 nuclear translocation, autophagy and protection from glucotoxicity depend on the chronic glycogen breakdown by glycogen phosphorylase.

HLH-30 has been shown to be required for the extension of lifespan promoted by longevity genetic mechanisms such as TOR inhibition, CR, mild mitochondrial metabolism impairment and downregulated insulin signaling[30,31]. However, the role of HLH-30 in the protection from glucotoxicity has not been previously addressed. While HLH-30 is critical to survive stresses such as starvation or pathogen

infection[37,38,43,59], our data demonstrates that its loss does not shorten the lifespan under excess glucose stress conditions in WT animals, but that HLH-30 expression is necessary for lifespan extension in *pgph-2 o/e* animals under glucotoxic conditions. This suggests that under excess glucose, WT animals rely on mechanisms independent of HLH-30 activation to ensure survival. In contrast to the results obtained with the loss of *hlh-30* in WT animals, loss of *atg-18* alone, a major component of the autophagy machinery, strongly decreased survival under glucotoxic conditions in comparison to WT animal, highlighting an important role for autophagy per se, and not HLH-30 per se, as a potent damage-repair or prevention mechanism under glucotoxicity. Thus, the diminution of lifespan due to glucotoxicity in WT animals appears to be independent of HLH-30, whereas the prolongation of lifespan promoted by *pgph-2 o/e* requires HLH30. In addition, our data indicate that both HLH-30 and autophagy are needed for the mode of action of PGPH-2 to enhance lifespan under glucotoxicity, but suggests the existence of HLH-30-independent autophagy activation mechanisms to promote longevity under excess glucose in WT animals.

We show that autophagy is particularly induced in the intestine of *pgph-2* overexpressing animals, but the type of activated autophagy (macroautophagy, lipophagy, aggrephagy, mitophagy, etc.) remains to be defined. Fat levels are strongly decreased in *pgph-2 o/e* animals, and RNA-seq experiments indicate that genes involved in lipid catabolism are constitutively induced, which may hint at an increased lipophagy by *pgph-2 o/e* but the contribution of lipophagy per se to fat loss by overexpression of *pgph-2* requires further exploration. Based on our previous work[7], the decreased fat accumulation in *pgph-2 o/e* animals is at least in part due to a decreased Gro3P-mediated lipogenesis rather than an increased lipolysis and triglycerides breakdown[7]. Other genetic and pharmacological interventions highlight also the relationship between fat levels and effects on lifespan and autophagy in *C. elegans*. Autophagy, and particularly lipophagy, has been clearly shown to play an important role in fat degradation. For instance, *eat-2* mutant animals[60,61], pharmacological or genetic AAK-2 (AMPK) activation[62–66], TOR RNAi[31,67], dietary restriction conditions display increased autophagy, decreased fat levels, and an extension of lifespan. However, while this sequence of processes has been observed in *C. elegans*, it did not always prove to be true, supporting the existence of additional regulatory events that link fat catabolism to autophagy and lifespan. For instance, *daf-2* mutant animals (Insulin-signaling-like receptor) display higher fat content than wild-type and also heightened autophagy and extended lifespan.

The role of glycogen in the maintenance of energy balance and the pathophysiology of aging has also recently received attention. Multiple studies in *C. elegans* have emphasized the distinct roles of glycogen reserves in stress resistance versus healthy longevity[68,69]. While glycogen increases survival, particularly under emergency conditions such as heat thermotolerance, oxidative or osmotic stress[69–72], its accumulation in normal aging has been shown to shorten lifespan, as reduced glycogen synthesis induced a shift from glycogen to trehalose and extended *C. elegans* lifespan[68,73]. In the present study, we

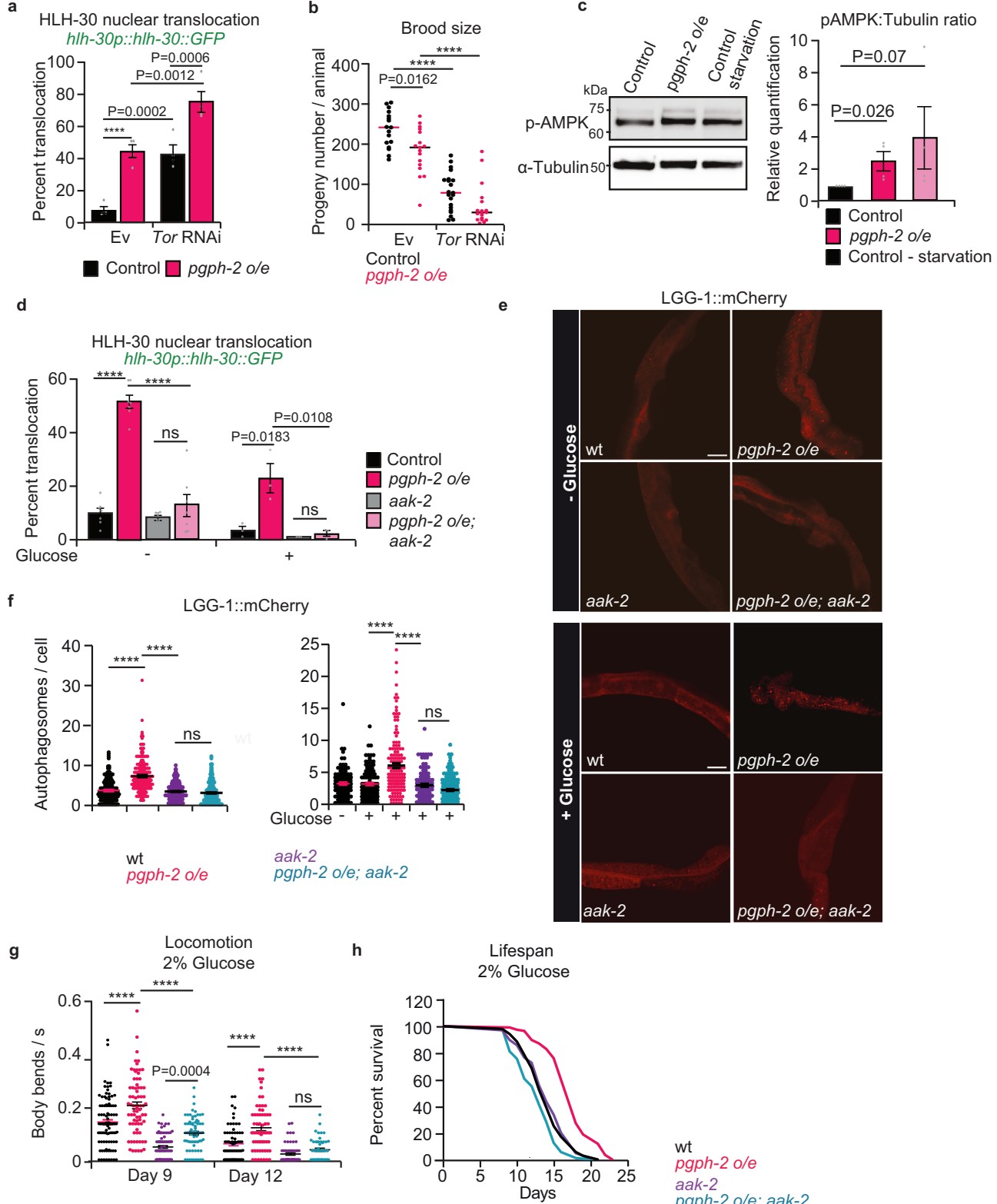

show that the activation of the PGPH-2 significantly reduces glycogen levels and that the inhibition of glycogen degradation fully abrogates the protection from glucotoxicity phenotype mediated by *pgph-2 o/e*.

What is the physiological relevance of increased levels of *pgph2* as it may relate to AMPK activity and CR? Noticeably, we show that *pgph-2 o/e* increases *eat-2* mRNA levels by twofold in conditions of excess glucose, which is in accordance with increased pharyngeal pumping observed in *pgph-2 o/e* animals under glucose conditions. Data from

previously published transcriptomics data[61,63] showed that both *eat-2* mutation (as a CR model) and activation of AAK-2, in fact, down-regulate *pgph-2* gene expression levels. This is physiologically relevant as *pgph2* o/e and the glycerol shunt act as a glucose excess detox-ification machine, and upon CR or enhanced AMPK activation, this pathway should be shut off.

The innate immune response genes are constitutively activated by *pgph-2* o/e, further supporting the CR-mimetic role of the glycerol

**Fig. 5 | PGPH-2 o/e activates the AMPK-HLH-30-autophagy signaling cascade to promote healthy aging under glucotoxicity.** For all figure panels, **** represents *P < 0.0001*. **a** Percent nuclear translocation of HLH-30 in young adult control animals and *pgph-2 o/e* animals on EV or TOR RNAi conditions. Data represent mean ± SEM from four independent experiments. *P*-values are obtained by one-way ANOVA with Bonferroni correction. **b** Brood size as measured by the number of progeny per animal in control and *pgph-2 o/e* animals grown on Ev plates or TOR RNAi plates. Data represent mean ± SEM from four independent experiments. *P*-values are obtained by one-way ANOVA with Bonferroni correction. **c** Western blot analysis and quantification by Image J of pAMPK and Tubulin in control, *pgph-2 o/e*, and starved control animals. Data represent mean ± SEM from four independent experiments. *P*-value is obtained by unpaired one-tailed Student's *t*-test. **d** Percent nuclear translocation of HLH-30 in young adult control, *pgph-2 o/e*, *pgph-2 o/e; aak-2 (ok524)*, and *aak-2 (ok524)* animals, grown on normal growth conditions or plates supplemented with 2% glucose. Data represent mean ± SEM from more than three

independent experiments. *P*-values are obtained by one-way ANOVA with the Bonferroni correction. **e, f** Representative confocal images (**e**) and quantification of LGG-1::mCherry puncta (**f**) in the intestines of day 1 WT, *pgph-2 o/e*, *pgph-2 o/e; aak-2 (ok524)*, and *aak-2 (ok524)* animals grown on normal and 2% glucose conditions. Data is shown using dot plots with denoted mean ± SEM from three independent experiments. *P*-values are obtained by one-way ANOVA with the Bonferroni correction. The scale bar = 20 μm. **g** Locomotion analysis on days 9 and 12 of age in control, *pgph-2 o/e*, *pgph-2 o/e; aak-2 (ok524)*, and *aak-2 (ok524)* strains grown on plates supplemented with 2% glucose. Data is shown using dot plots with denoted mean ± SEM from three independent experiments. *P*-values are obtained by one-way ANOVA with the Bonferroni correction. **h** Lifespan of control, *pgph-2 o/e*, *pgph-2 o/e; aak-2 (ok524)*, and *aak-2 (ok524)* overexpressing animals on plates supplemented with 2% glucose. The number of separate lifespan experiments, animals and detailed statistics are shown in Supplementary Data 1. *P*-values are obtained using the two-sided Mantel−Cox test.

shunt in promoting healthy aging. In accordance, recent advances highlight the role of CR in activating the innate immune response and the association between innate immunity and healthy longevity across taxa[20,39–42,74]. HLH-30/TFEB and DAF-16/FOXO have been demonstrated to act as combinatorial transcription factors to promote resistance to stress and extend lifespan[30]. Our data demonstrate that the healthy lifespan extension under excess glucose mediated by overexpression of *pgph-2* depends on HLH-30 and does not require DAF-16 nor EAT-2, suggesting that the activation of the glycerol shunt promotes longevity through a distinct non-canonical genetic pathway.

To our surprise, despite the strong depletion of fat and glycogen energy reserves in *pgph-2* overexpressing conditions, the AMP:ATP and ADP:ATP ratios were not significantly increased, suggesting that the animals may be relying on ATP-producing catabolic pathways other than lipolysis and glycogenolysis. The autophagy activation in *pgph-2* overexpressing conditions may explain this observation. In fact, lysosomes are known to contain various types of hydrolases, including proteases, lipases, glycosidases and the autophagic breakdown of macromolecules to maintain energy balance[75]. In support that the *pgph-2 o/e* and activation of the glycerol shunt does not lead to energy crisis, the levels of most TCA cycle intermediates, amino acid and nucleotides were not altered in *pgph-2 o/e* animals, and of interest, metabolites such as pyruvate, succinate and α-ketoglutarate were significantly heightened. In accordance, increased levels of pyruvate have been reported in CR *C. elegans* models[76], and the feeding of the worms with succinate, α-ketoglutarate and pyruvate has been shown to promote lifespan extension in *C. elegans*[76,77]. Recent work provided evidence that the mechanism of lifespan extension by α-ketoglutarate in *C. elegans* involved the coordination of the NAD+-SIRT1 signaling and peroxisomal function[78].

AMPK is a master regulator of energy homeostasis and the balance between AMPK activation and TOR inhibition orchestrates downstream metabolic signaling pathways that determine cellular fate[55,56]. Genetic or pharmacological AMPK activation or TOR inhibition has been shown to prolong healthy lifespan and increase resistance to various stresses in several model organisms[79–83]. In *C. elegans*, impaired glucose metabolism or glucose restriction activates AMPK via a mitohormesis, ROS-mediated pathway[84]. Also, recent work has identified a new AAK-2a isoform that functions exclusively in the neurons to mediate glucose-restriction-induced longevity, distinct from classical dietary restriction[85]. Our data show that the overexpression of PGPH-2 activates AAK-2 to increase healthspan and lifespan, particularly under glucotoxic condition. Importantly, strategies to activate AMPK to mimic starvation and activate lysosomal catabolic pathways are being explored in model organisms for translational benefits to humans. An aldolase inhibitor that reduces the catalysis of fructose-biphosphate to glyceraldehyde-3-phosphate, leading to reduced DHAP and Gro3P levels, has been recently shown to activate lysosomal AMPK and promote healthspan and lifespan both in mice and worms[86]. In

accordance, aldolase mutation in *C. elegans* which diminishes DHAP levels has been shown to restore the glucose-mediated short lifespan to normal while DHAP treatment greatly shortened lifespan[87]. Linking nutrient sensing to gene expression regulation, AMPK and mTOR regulate oppositely TFEB to rewire metabolism and induce the transcription of lysosomal, autophagy, and metabolic genes[32,43,51,59,88,89]. We showed that the nuclear translocation of HLH-30 by overexpression of PGPH-2 is not mediated by TOR suppression but rather requires AMPK activation to activate autophagy and promote a healthy lifespan under glucotoxic conditions.

The data presented in this study differ slightly from what we have previously reported using extrachromosomal array transgenes to overexpress PGPH-2[8]. While we have previously observed a very small but significant increase in lifespan under normal conditions and no reduction in brood size, our data here show that lifespan under normal conditions is unchanged and brood size is minimally but significantly decreased by PGPH-2 o/e. A stable and stronger expression of *pgph-2* in the integrated transgenic strains[7] likely explains the discrepancy in these phenotypes. Thus, with higher *pgph2* expression and glycerol shunt activation, it is anticipated that the animals will have a stronger depletion of energy stores, under normal conditions in comparison to excess glucose conditions. Particularly, fat and glycogen depots, which are critical for survival and reproduction, are strongly depleted in normal conditions and less reduced in glucotoxic conditions, which may explain the dichotomy in the outcome of *pgph-2 o/e* on lifespan. Also, the transcriptomics data hint at differences in *pgph-2o/e*-mediated gene expression regulation between normal and glucose excess conditions. RNA-seq volcano plots show a high number of differentially regulated genes in glucotoxic and not in normoxic conditions, and the heat map representations reveal that ROS detoxification and unfolded protein response pathways, for instance, are only differentially regulated by PGPH-2 o/e in glucotoxic and not in normoxic conditions.

CR has proven to be the most robust intervention that delays the onset of age-related diseases and compresses the years of decay in multiple organisms, including mammals and humans[11,42,79,83,90,91]. However, CR reduces fertility and is hard to pursue and achieve. Mimetics of CR that would promote the beneficial effects of CR without its associated caveats have clinical importance and are actively searched. A very attractive aspect of the glycerol shunt as a new target for healthspan is that the beneficial effects of PGPH-2 overexpression occurred at the same or even increased food intake in worms with unchanged[7] or mild reduction in fertility (current study).

It has become apparent that the AMPK-TFEB-autophagy axis is a main driver of healthy aging, particularly in CR conditions. From a therapeutic and drug discovery point of view, activators of AMPK, TFEB, and autophagy have been designed and are being actively considered to extend healthspan and to treat metabolic aging diseases, including neurodegenerative disorders. The finding reported in this study, that the enhanced activity of G3PP and of the glycerol shunt

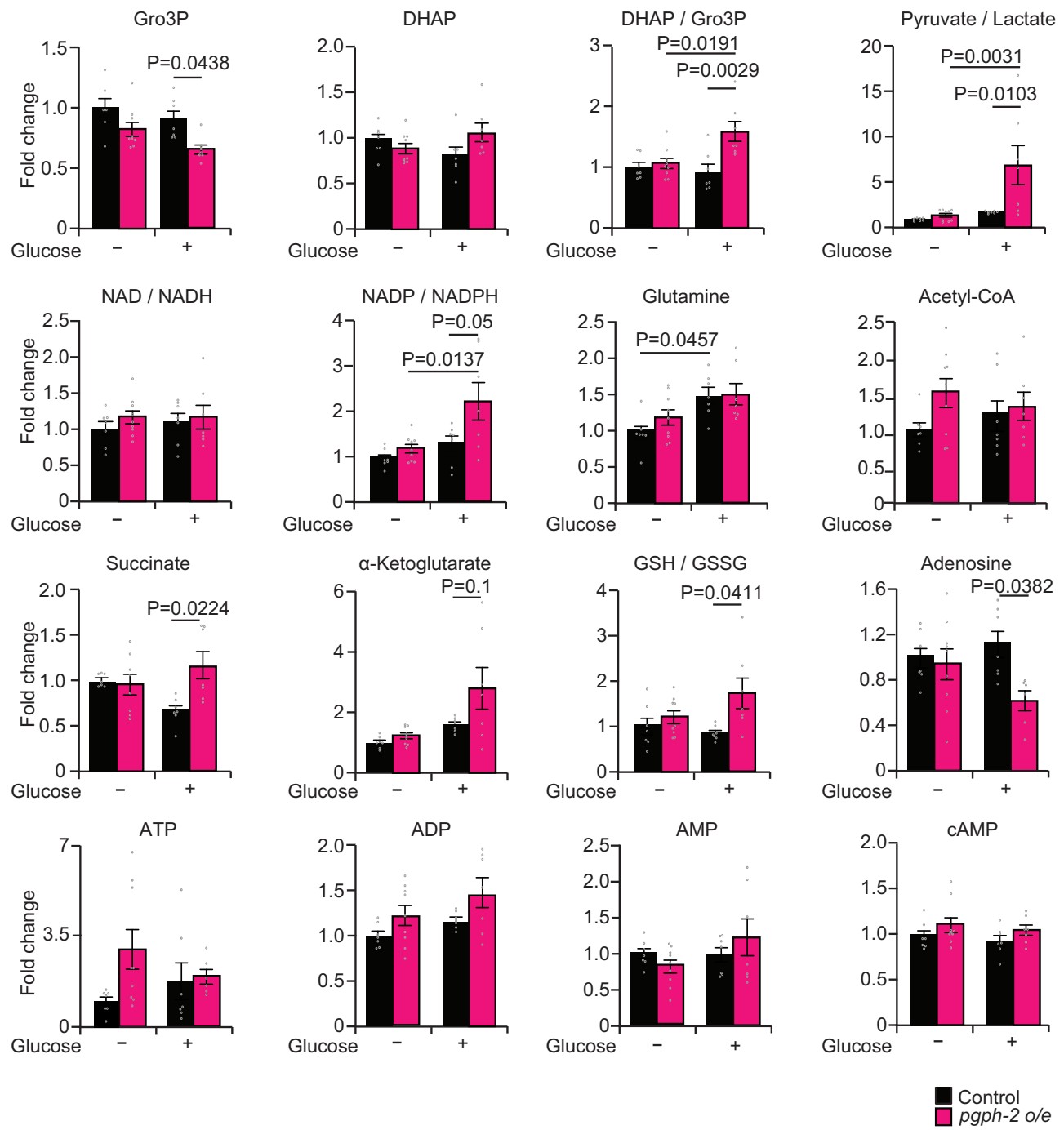

**Fig. 6 | Metabolomics analysis in _pgph-2 o/e_ animals.** Relative metabolite levels and ratios of some metabolites in synchronized L4/young adult control and _pgph-2 o/e_ animals grown on NGM plates or plates supplemented with 2% glucose. Data represent mean ± SEM from two independent experiments (control; _n_ = 7, _pgph-2 o/ e_; _n_ = 9, control–2% Glucose; _n_ = 7, _pgph-2 o/e_–2% Glucose; _n_ = 7). _P_-values are obtained by one-way ANOVA with the Bonferroni correction.

leads to constitutive activation of AMPK, TFEB, and autophagy, three key evolutionarily conserved longevity-promoting factors, sheds light on an unexplored role of G3PP and the glycerol shunt in the protection against aging-associated diseases and the promotion of healthy aging in disease contexts such as metabolic disorders associated with nutrient excess, atherosclerosis, and neurodegenerative disorders.

## Methods

### _C. elegans_ strains, maintenance and RNAi interference

Nematodes strains were maintained and synchronized using standard culture methods[92]. All experiments were conducted at 20 °C unless noted otherwise. Strains used in this study are available in

Supplementary Data 11. For normal growth conditions, OP50 _E. coli_ strain was used. For all RNAi experiments, phenotypes were scored with the F2 generation unless noted otherwise and HT115 bacteria transformed with empty vector were used as control.

### Synchronization methods

Two synchronization methods were used. For experiments that require a small number of progeny, including lifespan, locomotion, glucotoxicity, brood size, pharyngeal pumping, HLH-30 nuclear translocation and the confocal imaging of the autophagy reporter strains, animals were synchronized by transferring 5–8 gravid hermaphrodites to fresh agar plates to permit egg laying for few hours,

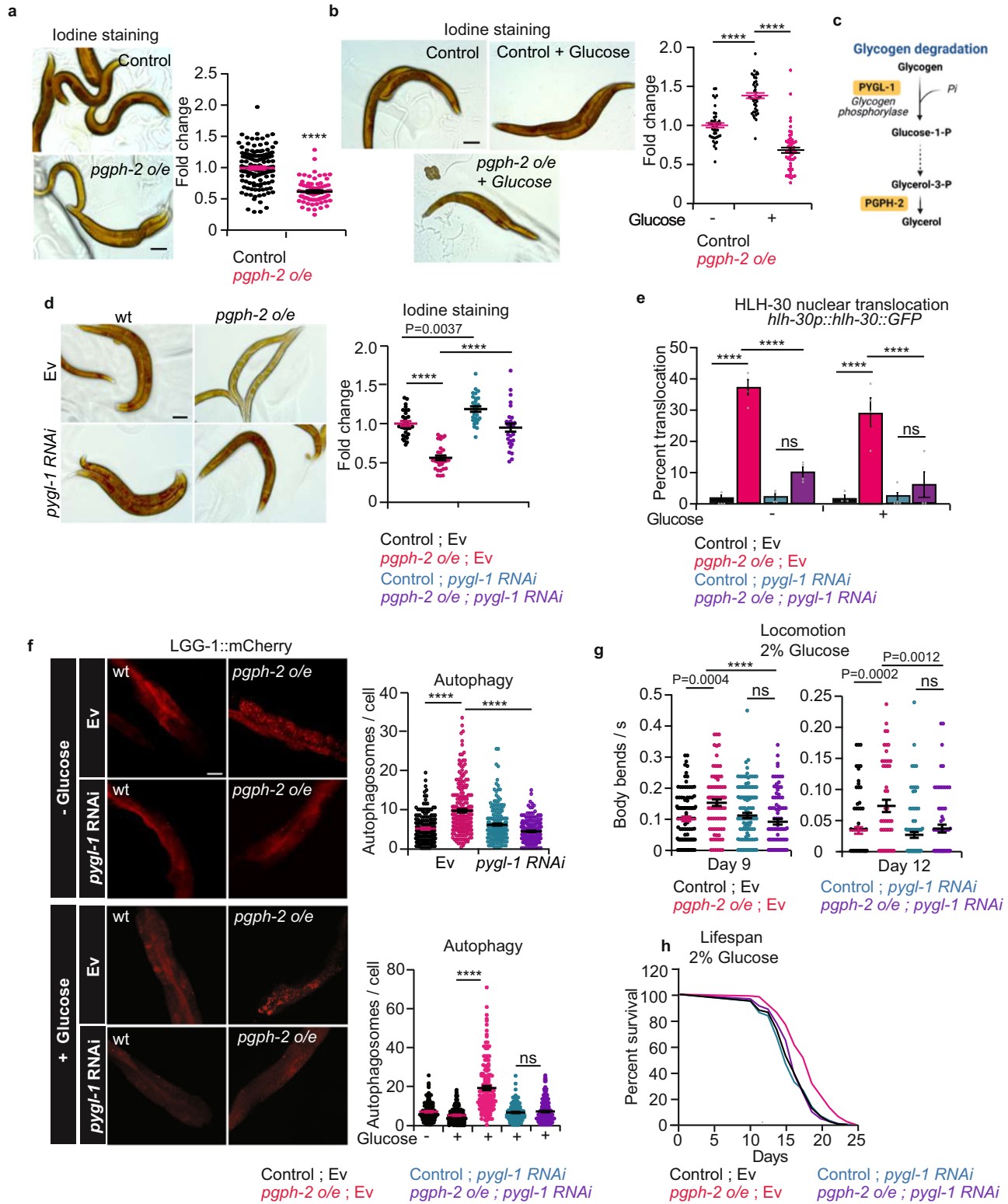

then the hermaphrodites were removed, and eggs were allowed to hatch and grow until they reached the experimental stage. For biochemical experiments, including RNA extraction and sequencing and metabolite measurement experiments, the nematodes were synchronized by the standard hypochlorite bleaching method[93].

### Construction of the stable transgenic *pgph-2 o/e* strain
The overexpressing *pgph-2* plasmids were generated using Clontech In-Fusion PCR Cloning Kit according to the manufacturer's

protocol[94]. The promoter region and genes were amplified from N2 worm genomic DNA and were cloned into ppD49.26 vector using *sbf-1* and *kpn-1* cloning sites. Cloning mixtures were transformed using stellar competent cells according to the manufacturer's protocol. Then, 10 ng/μl of plasmid DNA, 20 ng/μl mCherry, and 170 ng/μl pbluescript were injected into worm gonad arms of WT worms, and the positive lines were maintained after transmission to the F3 generation. The transgene was then integrated using UV irradiation according to protocol[24].

**Fig. 7 | PGPH-2 o/e reduces glycogen stores to activate an AMPK-dependent/ HLH-30/autophagy cascade.** For all figure panels, **** represents *P < 0.0001*. **a**, **b** Representative iodine staining images and quantification in control and *pgph-2* o/e animals in normal growth conditions (**a**) and plates supplemented with 2% glucose (**b**). The scale bar = 50 μm. Data represent mean ± SEM from three independent repeats. *P*-values are obtained using two-sided unpaired Student's *t*-test (**a**) and one-way ANOVA with Bonferroni correction (**b**). The scale bar = 50 μm. **c** Glycogen degradation pathway. **d** Representative iodine staining images and quantification in WT and *pgph-2* o/e exposed to Ev and *pygl-1* RNAi. Data represent mean ± SEM from three independent repeats. *P*-values are obtained using one-way ANOVA with Bonferroni correction. **e** HLH-30 nuclear translocation in WT and *pgph-2* o/e animals expressing the HLH-30::GFP transgene exposed to Ev and *pygl-1* RNAi for two generations and grown on normal conditions or excess glucose conditions. Data represent mean ± SEM from three independent experiments.

*P*-values are obtained using one-way ANOVA with Bonferroni correction. **f** Representative confocal images and quantification of LGG-1::mCherry puncta in the intestines of day 1 WT and *pgph-2* o/e exposed to Ev and *pygl-1* RNAi for two generations and grown on normal or excess glucose conditions. Data represent mean ± SEM from three independent experiments. *P*-values are obtained using one-way ANOVA with Bonferroni correction. The scale bar = 20 μm. **g** Locomotion analysis on days 9 and 12 of age, measured by body bends per second in control and *pgph-2* o/e exposed to Ev and *pygl-1* RNAi bacteria and grown on plates supplemented with 2% glucose. Data is shown using dot plots with denoted mean ± SEM from three independent experiments. *P*-values are obtained using one-way ANOVA with the Bonferroni correction and exact number of tracks is indicated in datasource. **h** Lifespan of control and *pgph-2* o/e exposed to Ev and *pygl-1* RNAi bacteria and grown on plates supplemented with 2% glucose. *P*-value is obtained using the two-sided Mantel−Cox test.

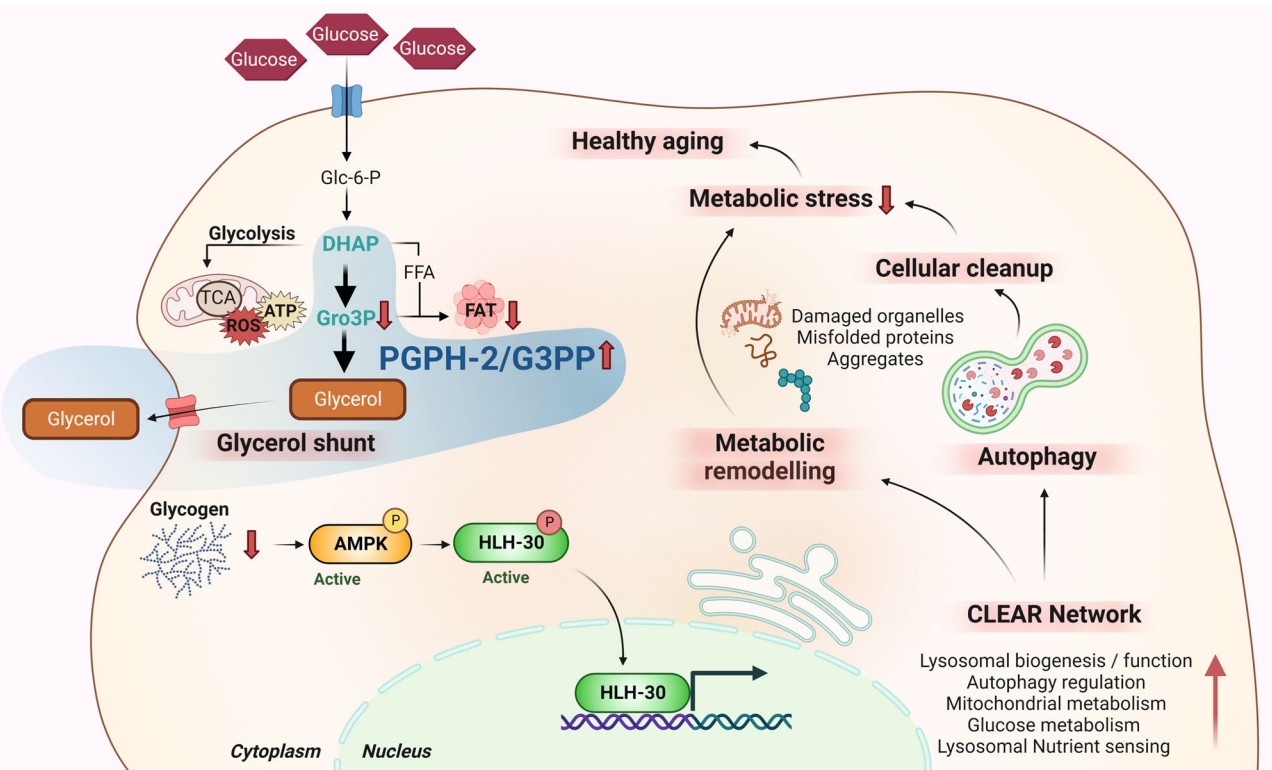

**Fig. 8 | Graphical representation of the study findings.** In this work, we unveil the mechanism whereby G3PP activation mimics in part the beneficial effects of calorie restriction, without the caveats of calorie intake restriction and significant reduced fertility, to promote healthy aging and lifespan extension, particularly in glucose-excess conditions. Specifically, we demonstrate that the overexpression of the major worm G3PP homolog, PGPH-2, activates the AMPK-TFEB/HLH-30-autophagy axis, three key longevity factors and attractive therapeutic venues to treat metabolic diseases that appear with age. In addition, we show that the initial step that triggers the activation of this signaling cascade is the chronic depletion of glycogen stores mediated by *pgph-2 o/e*. Figure created using Biorender.com.

## Lifespan and glucose toxicity assays

All lifespan and glucose toxicity curves were performed at 20 °C. Briefly, *C. elegans* nematodes were synchronized as described above. For glucose toxicity, the animals were constantly transferred to *E. coli*-seeded agar plates supplemented with 2% glucose. Three days after egg preparation, around 120 animals were manually transferred (40 animals per plate in triplicates) to fresh plates. Worms were transferred daily for the first week and every other day afterward until the aged population was well separated from growing larvae. Worms were scored daily and were considered alive if they responded to gentle tapping with a platinum wire. Worms that crawled off the plate or died from bagging or internal hatching were censored from the analysis.

## Locomotion assays

Along with the lifespan and glucose toxicity assays, the locomotion behaviors of the nematodes were filmed for 30 seconds at days 9 and

12 from adulthood, unless indicated otherwise, using the Wormlab-system (MBF Bioscience), version 2018. Worms on the plates were tracked and the number of body bends was manually assessed and divided by the time of the movie clip.

## Pharyngeal pumping

Pharyngeal pumping rates were measured by counting the number of grinder movements per 30 seconds using a stereomicroscope at indicated age, which is days from adulthood and not days from birth. The worms were always on a lawn of food at rest. Pharyngeal pumping rates were measured in at least 10 worms per condition for every independent repeat.

## Brood size

Single L4 animals were transferred to fresh agar plates seeded with *E. coli* OP50 bacteria. The animals were transferred individually to fresh

plates every day until they stopped laying eggs after 5–6 days. The progeny number was counted 2 days after egg laying, and the brood size was determined by the sum of the number of progeny produced by individual hermaphrodite.

## Autophagy counts in intestinal cells

Worms were synchronized on standard NGM plates until the young adult stage was reached. On the day of the experiment, 15–20 animals were mounted on 2% agarose slides and immobilized with 5 mM Levamisole dissolved in M9 buffer. In vivo confocal fluorescent imaging was then performed using a Leica confocal microscope, and images were further analyzed. Intestinal cells were then delineated, and the number of LGG-1::mCherry autophagosomes per intestinal cell was counted by three independent researchers and the average of the counts was calculated. Experiments were repeated independently more than three times.

## HLH-30 nuclear translocation

The nuclear localization of HLH-30::GFP was visualized with a Leica fluorescence dissecting microscope. About 30 animals were used per condition, and the percentage of animals showing the nuclear translocation was counted.

## RNA extraction and purification

Synchronized young adult nematodes were harvested, washed with M9, and flash-frozen in liquid nitrogen. RNA was extracted using Trizol and purified using QIAGEN RNeasy columns. RNA samples were processed for RNA-seq analysis at Novogene Inc.

## RNA-seq library preparation

A total amount of 1 μg RNA per sample was used as input material for the RNA sample preparations. Sequencing libraries were generated using NEBNext® Ultra™ RNA Library Prep Kit for lliumina® (NEB, USA) following the manufacturer's recommendations, and index codes were added to attribute sequences to each sample. Briefly, mRNA was purified from total RNA using poly-T oligo-attached magnetic beads. Fragmentation was carried out using divalent cations under elevated temperature in NEBNext First Strand Synthesis Reaction Buffer (5X) or by using sonication with Diagenode bioruptor Pico for breaking RNA strands. First-strand cDNA was synthesized using a random hexamer primer and M-MuLV Reverse Transcriptase (RNase H). Second-strand cDNA synthesis was subsequently performed using DNA Polymerase I and RNase H. Remaining overhangs were converted into blunt ends via exonuclease/polymerase activities. After adenylation of the 3′ ends of DNA fragments, NEBNext Adaptor with hairpin loop structure was ligated to prepare for hybridization. In order to select cDNA fragments of preferentially 150–200 bp in length, the library fragments were purified with the AMPure XP system (Beckman Coulter, Beverly, USA). Then 3 μl USER Enzyme (NEB, USA) was used with size-selected, adapter-ligated cDNA at 37 °C for 15 min followed by 5 min at 95 °C before PCR. Then PCR was performed with Phusion High-Fidelity DNA polymerase, Universal PCR primers and Index (X) Primer. At last, PCR products were purified (AMPure XP system), and library quality was assessed on the Agilent Bioanalyzer 2100 system. The clustering of the index-coded samples was performed on a cBot Cluster Generation System using PE Cluster Kit cBot-HS (Illumina) according to the manufacturer's instructions. After cluster generation, the library preparations were sequenced on an Illumina platform, and paired-end reads were generated.

## RNA-seq data analysis

Raw data (raw reads) of fastq format were first processed through in-house perl scripts. In this step, clean data (clean reads) were obtained by removing reads containing adapter, reads containing ploy-N and low-quality reads from raw data. At the same time, Q20, Q30 and GC content the clean data were calculated. All the downstream analyses were based on the clean data with high quality. Reference genome and gene model annotation files were downloaded from the genome website directly. Index of the reference genome was built using Hisat2 v2.0.5, and paired-end clean reads were aligned to the reference genome using Hisat2 v2.0.5. We selected Hisat2 as the mapping tool for that Hisat2 can generate a database of splice junctions based on the gene model annotation file and thus a better mapping result than other non-splice mapping tools. The mapped reads of each sample were assembled by StringTie (v1.3.3b) in a reference-based approach. Feature Counts v1.5.0-p3 was used to count the reads numbers mapped to each gene. And then FPKM of each gene was calculated based on the length of the gene and reads count mapped to this gene. FPKM, the expected number of Fragments Per Kilobase of transcript sequence per Millions base pairs sequenced, considers the effect of sequencing depth and gene length for the reads count at the same time and is currently the most commonly used method for estimating gene expression levels. We used four biological replicates per condition with eight groups. With 32 samples, four replicates per condition for eight conditions, assuming a medium effect size of 0.25, there is 83.7% power for rejecting the null hypothesis when, in fact, it is false (avoid Type II error). This percentage falls within the acceptable norm (80–90%)[95,96]. Differential expression analysis of two conditions/groups (four biological replicates per condition) was performed using the DESeq2 R package (1.20.0). DESeq2 provides statistical routines for determining differential expression in digital gene expression data using a model based on the negative binomial distribution. The resulting $P$-values were adjusted using Benjamini and Hochberg's approach for controlling the false discovery rate. Genes with an adjusted $P$-value ≤0.05 found by DESeq2 were assigned as differentially expressed. Gene ontology analysis was performed using g:Profiler. For all the software names and versions used for RNA-seq data processing, please see Supplementary Data 12.

## Oil red O staining

Oil red O staining was performed as previously described with small modifications[59]. Briefly, synchronized young adult animals were collected with M9 buffer and washed twice with PBS1x-0.01% triton X. Animal pellets were treated for 15 min with PBS1x, 0.01% triton X, 60% isopropanol solution and transferred to Eppendorff tubes and incubated in 60% oil red O solution overnight. The oil red O stock solution was equilibrated for more than 3 days, and the 60% diluted solution was equilibrated overnight and filtered two times on the day of the experiment. Images were taken using a Leica microscope and were analyzed using image J.

## Immunoblotting

For the phospho-AMPK immunoblotting, synchronized young adult animals were harvested and washed three times with M9 buffer and pellets were flash-frozen in liquid nitrogen. Pellets were then lysed in the AMPK lysis buffer (50 mM Tris pH8, 150 mM NaCl, 1% NP-40, 0.1% SDS, 10 mM EDTA, 10 mM HEPES, 50 mM potassium acetate, 2.5 mM sodium acetate, 1 mM magnesium acetate, 0.5 mM EGTA, 5 mM MgCl₂, 0.25 mM DTT, 15 μg/μl digitonin) supplemented with 1 μM PMSF and phosphatase inhibitors, 2 mM Na₃VO₄ and 1 mM NAF. Samples were sonicated for 10 min with intermittent ice incubation, and proteins were separated on SDS-PAGE gels and revealed by western blot using the pAMPKα (T172) rabbit mAb from Cell Signalling (catalog number: 2535S; 1:1000 dilution prepared in TBS-T1x+5% BSA + 50 μM NaF and incubated overnight at 4 °C). The α-tubulin antibody is from Abcam (catalog number: ab4074; 1:10000 dilution prepared in 5% milk in TBST1X). Uncropped western blots of Fig. 5c are shown in datasource.

## Metabolite extraction

Glucose treatment: Synchronized L1 animals by sodium hypochlorite were plated on NGM plates or plates supplemented with 2% glucose for

48 h, then harvested with M9 buffer and washed three times before flash freezing in liquid N2. Metabolites were analyzed by LC-MS/MS[6] with small modifications. Worm pellets (~200 mL) were collected in CK14−2 mL lysing kit tubes (Bertin Technologies), flash-frozen and stored at −80 °C. Samples were slowly thawed and homogenized using 4.25 volumes of ice-cold aqueous methanol 98.8%, 2.4 mM ammonium acetate (pH 9), containing 10 mM ($^{13}C_{10}$,$^{15}N_5$)-AMP as internal standard, using bead beating (Precellys plus Cryolys cooler, Bertin Technologies; protocol: 6000 rpm, 2 × 25 s, 15 s pause) and 2 min sonication (cycles of 10 s on/off, output of 150 W) in a cup-horn sonicator filled with water and ice. Homogenates were incubated for 15 min on ice and centrifuged at 20,000×*g*, 15 min at 4 °C. Water-soluble metabolites were then extracted from the supernatant by liquid-liquid extraction and analyzed as described[6]. Relative amounts were normalized to protein content using BCA.

## Statistical analyses

Data are expressed as means ± SEM. Statistical analyses were performed by Student's *t*-test for two groups or one-way ANOVA for multiple groups using GraphPad. For survival curves, we used the Log-rank Mantel−Cox test. Significance is indicated in the figures and legends (*$P < 0.05$, **$P < 0.01$, ***$P < 0.001$, ****$P < 0.0001$) or included in the supplementary datasets.

## Reporting summary

Further information on research design is available in the Nature Portfolio Reporting Summary linked to this article.

## Data availability

All relevant data generated or analyzed during this study are included in this manuscript and/or its supplementary information, or can be obtained from the corresponding authors upon request. The data underlying Figs. 1–7 and Supplementary Figs. 1–7 are provided as Source data. The metabolomics data is now publicly available at Metabolights MTBLS6847 and the RNA-seq at GEO GSE228784. Source data are provided with this paper.

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

## Acknowledgements

This study was supported by funds from the Canadian Institutes of Health Research (143308, to M.P. and S.R.M.M.), and from Dasman Diabetes Research Institute/ Montreal Medical International (to M.P., S.R.M.M., R.A. and F.A.-M.). M.P. was a recipient of a Canada Research Chair in Diabetes and Metabolism up to 2019. E.P. was a recipient of a postdoctoral fellowship from Diabetes Canada up to 2021. L.C. was supported by UdeM PREMIER and H.S. by a summer fellowship from Diabetes Canada. We thank the metabolomics and imaging facilities of CRCHUM. We acknowledge the *Caenorhabditis* Genetic Center for *C. elegans* strains. We also gratefully acknowledge Drs. Siegfried Hekimi and Richard Roy for helpful discussions and reagents. We also thank the Prentki lab members, particularly Drs. Marie-Line Peyot and Pegah Poursharifi, for thoughtful discussions. We also acknowledge Dr. Parker's lab members, specifically Dr. Audrey Labarre and Gilles Tossing, for their help and support.

## Author contributions

E.P., A.P., S.R.M.M., and M.P. contributed to the research design. E.P., L.-L.K., R.Z., L.C., P.S., Y.B., H.S., A.A., and A.O., contributed to experiment performances. E.P., D.Z., R.A., A.P., S.R.M.M., F.A.-M., and M.P. contributed to the analysis of the data and editing of the manuscript, and E.P. and M.P. contributed to the writing of the paper.

## Competing interests

The authors declare no competing interests.
