## [Peer Review File · Nature Communications]

Glycerol 3-phosphate phosphatase/PGPH-2 counters metabolic stress and promotes healthy aging via a glycogen sensing-AMPK-HLH-30-autophagy axis in *C. elegans*REVIEWER COMMENTS

Reviewer #1 (Remarks to the Author):

In this manuscript entitled “Glycerol 3-phosphate phosphatase/PGPH-2 counters metabolic stress and promotes healthy aging via a glycogen sensing-AMPK-HLH-30-autophagy axis in *C. elegans*”, the authors further determined how PGPH-2 increased longevity under glucotoxicity conditions, as a followup of their previous paper (PMID: 35017476). The authors showed that integrated *pgph-2* transgenes caused physiological effects similar to those of calorie restriction, including increased lifespan, reduced fat accumulation, and improved locomotion, in addition to protecting worms from glucotoxicity. The authors then found that PGPH-2 overexpression increased healthspan and lifespan in an HLH-30-dependent manner. The authors found that glycogen reduction by *pgph-2* overexpression enhanced AMPK signaling, leading to upregulation of HLH-30-mediated autophagy. They also included RNA seq and metabolomic analysis upon overexpressing *pgph-2*. Overall, this study is a nice followup of their recent paper in *Nat. Comm.* (PMID: 35017476). However, I think the information this paper add to the research field is incremental and do not believe the paper provides a conceptual leap required for *Nat. Comm.* Following are my specific concerns that the authors should address.

Major concerns

1. This paper has many data but the description of the data and the discussion is very loose. For example, it is not clear what is new and what is confirmation for this paper compared to the authors' previous *Nat. Comm.* paper. In addition, previous papers that are very relevant to this paper's data were not cited or discussed. For example, their data regarding glycogen should be discussed with respect to a very relevant paper (PMID: 28627510). Their data about DHAP level should be discussed with another very relevant previous paper (PMID: 26637528). Their data about AMPK should be discussed with one of the pioneering papers regarding the glucose on *C. elegans* lifespan (PMID: 17908557). These are just some examples that I easily found and many important papers are missing in the references as well. The fact that none of these key papers were not even cited is very disappointing. I strongly suggest that the authors thoroughly do literature search and revise the manuscript based on that.
2. The RNA seq and metabolomic data are not tied together with other parts and should be analyzed again as the parts of one paper.

Minor concerns

1. Please show appropriate images of Oil red O staining or fluorescent signal that match the quantification data.
2. Please consider changing figure 6 to supplementary figures.
3. Ultimately, *pygl-1* RNAi treatment is an indirect way to glycerol-generating pathway. I recommend that the authors should use *pgph-2* mutants to strengthen their results.

4. In figure 1, it may be more effective to combine panels D and F to show the improved locomotion of *pgph-2* o/e worms.
5. In figure 1H, it will be better to show lifespan assay data comparing *eat-2* worms and *pgph-2* o/e; *eat-2* worms as the authors did in figure 1J.
6. In figure 1H, does *pgph2* o/e have effects on brood size under dietary restriction conditions?
7. In figure 2B, please add images for high-glucose conditions.
8. Do mutants with decreased fat levels display similar effects on lifespan or autophagy like *pgph2* o/e?
9. In figure 3A, please use a higher-resolution image for the scheme.
10. In figure 3B, it will be better to change labeling with number to 'upregulated' and 'downregulated'.
11. Please doublecheck positioning of panel legends and consistency of legend styles.

Reviewer #2 (Remarks to the Author):

This manuscript showed PGPH-2 overexpression decreases fat accumulation, increases healthspan measured by pharyngeal pumping and locomotion, and extends lifespan under high glucose. The authors further identified TFEB/HLH-30 transcription factor activation, autophagy, AMPK activation, and glycogen storage reduction are required for PGPH-2 to benefit health. They proposed a pathway of PGPH-2—glycogen storage—AMPK—HLH-30—autophagy for the health benefits they examined in this study. The authors also analyzed gene expression and metabolites changed by PGPH-2 overexpression via RNA-seq and metabolomics.

The findings are novel and of interest to the fields of aging. Most of the conclusions were well supported by experiments. I have several comments:

1. A primary concern is the physiological relevance of PGPH-2 overexpression. The study throughout this manuscript was performed using PGPH-2 overexpression transgenic strain. And they propose PGPH-2 overexpression as a CR mimetic. It would strengthen the manuscript by showing the physiological significance of increased level of PGPH-2, if the authors test whether CR induces PGPH-2 levels (RT-PCR, fluorescent reporter, or western blot). Similar, whether AMPK activator induces PGPH-2 levels.
2. PGPH-2 OE is stated as a CR mimetic in the manuscript, but it can't extend lifespan under normal glucose condition like CR. The statement of PGPH-2 OE as a CR mimetic needs to be amended.
3. I would suggest more experiments to describe the healthspan and lifespan discrepancies under normal and glucotoxicity condition. For example, is PGPH-2-glycogen storage-AMPK-HLH-30-autophagy pathway only activated under glucose access condition? Does *hlh-30*, *aak2*, *atg-18* mutant, or *pygl-1* RNAi blunt locomotion in normal condition (without 2% glucose)? The involvement of glycogen storage by *pygl-1*

RNAi is examined under normal condition (without 2% glucose) for HLH-30 nuclear translocation and autophagy. Does *pygl-1* RNAi also abolish HLH-30 and autophagy activation under 2% glucose? Why RNA-seq pathways are similar between PGPH-2 OE with or without 2% glucose, however the lifespan outcomes are different? Why autophagy and *hlh-30* nuclear translocation are activated in an AMPK dependent way with or without glucose, however the lifespan outcomes are different? Is PGPH-2 only beneficial under glucotoxicity but not normal condition? These questions need to be addressed experimentally or textually. Authors' thoughts on whether PGPH-2 is a beneficial gene like the statement of "CR mimetic" or PGPH-2 is only important under glucotoxicity should be discussed.

4. Fig. 2C, *hlh-30* seems to be significantly lower in body bends compared to WT, how do you know it is just sick, and PGPH-2 can't reverse it? Similarly, in Fig. 4D and E, same question in terms of *atg-18*, *atg-18* KO has lower motility and survival rates in lifespans. I think proper controls need to be shown when a manipulation affects the phenotype already when compared to control. This needs to be addressed experimentally or textually.

5. Fig. 5C westernblot result is not solid. AMPK protein level needs to be shown. Quantification and statistics need to be shown based on p-AMPK/AMPK ratio.

6. Authors showed RT-PCR results of *Igg-1*, *Igg-2*, *atg-18* under PGPH-2 OE in Fig. S4A. Any autophagy genes including these three were revealed in the RNA-seq data? This needs to be described and discussed.

Minor points:

Fig. 1A,B,C, Fig. 5A, Fig. S1A: please use dot scored plots.

Fig. 1D, F, H: please use a different color for error bars of control and PGPH-2 OE, it confuses people by use the reciprocal color of the comparing condition.

Page 6, "with a marginal effect on lifespan (Figure 1D-E)". please add "non-significant" or describe it as no effect.

Page 6, "and also medium and maximal lifespan under glucotoxicity". I believe "increase" is missing in the sentence.

Page 6, "In addition, the newly generated transgenic line phenocopies what we have observed using extrachromosomal arrays [7]". Please rephrase this to be clearer to a non-worm reader so they know what the difference is here between the transgenic line and extrachromosomal arrays generated lines.

Page 6, "Noticeably, the brood size decrease by PGPH-2 o/e is minimal in comparison to the severe brood size decrease observed in *eat-2* animals in normal growth and excess glucose conditions (Figure S1D)". What does this sentence mean in line with the conclusion of this panel? The use of "minimal" is confusing.

Page 7, "not due to a transgene artefact", artefact is a typo.

Page 11, "preventing the transcription of the CLEAR network," Please spell out CLEAR.

Fig. S5B and C, needs quantification and statistics.

Fig. 5B, color legend in the figure is missing.

Reviewer #3 (Remarks to the Author):

Manuscript No: NCOMMS-22-47059

Glycerol 3-phosphate phosphatase/PGPH-2 counters metabolic stress and promotes healthy aging via a glycogen sensing-AMPK-HLH-30-autophagy axis in *C. elegans*

This manuscript by Possik *et al.* reports critical insights into the role of G3PP/PGPH-2 in high glucose diet in *Caenorhabditis elegans*. The authors demonstrate that the overexpression of *pgph-2* activates a cascade of events on the AMPK-TFEB-autophagy axis that might serve as an alternative to healthy ageing by overcoming the side effects of calorie restriction, such as reduced fertility. The manuscript presents answers to scientifically relevant hypotheses supported by clear, compelling figures. However, some sections should be reconsidered to alleviate minor concerns—outlined below—which could be resolved by reanalysing parts of the data.

Comments:

- The methods section for bioinformatic analysis states that the authors used the *featureCounts* function (from *Rsubread* package) to obtain gene counts. It is not clear whether the raw data was mapped or aligned to the reference genome before feature counting or skipped alignment using tools such as *salmon* that is bias aware (with *--gcBias* parameter). The reference genome version should also be added (i.e., GenBank or RefSeq assembly accession) for either method.

- The authors used *DESeq2* package for conditions/treatments with biological replicates and *edgeR* package for conditions without biological samples to analyse RNA-seq data. Although these choices were appropriate analysis methods, the type of input data (RPKM) is not appropriate for the *DESeq2* method (See original paper PMID: 25516281). First, RPKM should be used with single-end reads, while FPKM is used for paired-end RNA-seq data (though the legend for Figure S2A states FPKM was used). Given the authors analysed paired-end data, the usage of FPKM would be more accurate (See PMIDs: 32284352; 34158060). Second, despite their popularity, these units (i.e., RPKM,

FPKM and TPM) should rather be used for qualitative assessments such as Principal Component Analysis (PCA) but not for differential expression analysis.

- The authors analysed four biological replicates per condition for RNA-seq experiments (Figure S2)– though the methods state ‘three’ replicates. The reproducibility problem in *C. elegans* research has been addressed in several reviews, including Urban *et al.* (2021) [PMID: 34793939] and Pho & MacNeil (2019) [PMID: 31127069]. For instance, two out of four replicates in the *hlh-30 + Glucose* condition have differential R^2 values than the other two replicates (Figure S2B). Therefore, it would be more convincing and conclusive if the authors computed power analysis to show the number of replicates per condition was robust enough. Also, a 2-dimensional PCA plot would be easier to comprehend the variation within a condition than the 3-dimensional PCA plot (Figure S2C) since the third principal component is rather low (5.45%).

Other recommended refinements:

- *Reporting of statistics:* The authors reported experimental results with transparency (visible individual data points) and the appropriate type of statistical tests. Findings were visualised by using average values and standard error of the means (mean \pm SEM). Although mean \pm SEM continues to be reported in life sciences research, SEM does not describe the sample. Therefore, the mean values should be reported with standard deviation (SD), not SEM (PMIDs: 33402813, 23125963). Given the robustness of the experimental procedures and results, changing SEM to SD will not affect their outcomes. Nevertheless, it would improve the quality of the manuscript from a statistics point of view.

- Adding software versions and data availability could improve the reproducibility of the analysis.

- The data source for Figure 1F (the exact number of tracks) is missing.

- The legend of Figure 7 for some of the subplots is mislabelled/missing:

Original legend:

“Figure 7: PGPH-2 o/e reduces glycogen stores to activate an AMPK-dependent/HLH- 30/autophagy cascade. A-B) Representative iodine staining images and quantification in control and *pgph-2* o/e animals in normal growth conditions (A) and plates supplemented with 2% glucose (B). (C) HLH-30 nuclear translocation in WT and *pgph-2* o/e animals expressing the HLH-30::GFP transgene exposed to Ev and *pygl-1* RNAi for two generations. Data represent mean \pm SEM from three independent experiments. (D) Representative confocal images and quantification of LGG-1::mCherry puncta in the intestines of day 1 WT and *pgph-2* o/e exposed to Ev and *pygl-1* RNAi for two generations. Data represent mean \pm SEM from three independent experiments. The scale bar

indicates 20 μm . (E-F) Locomotion analysis on days 9 and 12 of age, measured by body bend per second in control and *pgph-2* o/e exposed to Ev and *pygl-1* RNAi bacteria and grown on plates supplemented with 2% glucose. Data is shown using dot plots with denoted mean \pm SEM from three independent experiments. P-values are obtained by one-way ANOVA with the Bonferroni correction. E) Lifespan of control and *pgph-2* o/e exposed to Ev and *pygl-1* RNAi bacteria and grown on plates supplemented with 2% glucose. The number of separate lifespan experiments, animals and detailed statistics are shown in Supplementary Table S1. P-value is obtained using the two-sided Mantel-Cox test.”

Suggested legend:

“Figure 7: PGPH-2 o/e reduces glycogen stores to activate an AMPK-dependent/HLH-30/autophagy cascade. A-B) Representative iodine staining images and quantification in control and *pgph-2* o/e animals in normal growth conditions (A) and plates supplemented with 2% glucose (B). C) Glycogen degradation pathway. D) Representative iodine staining images and quantification in WT and *pgph-2* o/e exposed to Ev and *pygl-1* RNAi in {XXX} conditions. E) HLH-30 nuclear translocation in WT and *pgph-2* o/e animals expressing the HLH-30::GFP transgene exposed to Ev and *pygl-1* RNAi for two generations. Data represent mean \pm SEM from three independent experiments. F) Representative confocal images and quantification of LGG-1::mCherry puncta in the intestines of day 1 WT and *pgph-2* o/e exposed to Ev and *pygl-1* RNAi for two generations. Data represent mean \pm SEM from three independent experiments. The scale bar indicates 20 μm . G) Locomotion analysis on days 9 and 12 of age, measured by body bends per second in control and *pgph-2* o/e exposed to Ev and *pygl-1* RNAi bacteria and grown on plates supplemented with 2% glucose. Data is shown using dot plots with denoted mean \pm SEM from three independent experiments. P-values are obtained by one-way ANOVA with the Bonferroni correction. H) Lifespan of control and *pgph-2* o/e exposed to Ev and *pygl-1* RNAi bacteria and grown on plates supplemented with 2% glucose.”

The manuscript addresses the role of *pgph-2* on the lifespan and healthy ageing through the enhanced activity of PGPH-2 and of the glycerol shunt which leads to constitutive activation of AMPK, HLH-30 and autophagy, mimicking only beneficial effects of calorie restriction. Findings on the unexplored role of PGPH-2 and the glycerol shunt demonstrate protection against ageing and promote fit and healthy longevity. This study by Possik *et al.* will contribute indispensable discoveries to ageing research on *C. elegans* literature.

Point by point response to reviewers, manuscript: NCOMMS-22-47059

We thank the reviewers for their valuable and helpful comments and suggestions. This helped us to improve the manuscript and clarify certain issues that we either missed or did not explain well. We addressed to the best of our abilities the vast majority of the major and minor concerns by performing many additional experiments. Thus, 12 additional figure panels or sub panels and two additional supplementary tables have been added to the original version of the figures. For a few suggested experiments that cannot be performed for technical reasons, we provide an explanation.

The figure legends and datasource tables have also been revised according to the modification in the manuscript.

The manuscript text, figures, tables, supplementary information and referencing have been formatted according to the *Nature Communications* guidelines. Also the metabolomics and transcriptomics have been submitted to the Metabolights database (MTBLS6847) and GEO repository (GSE228784), respectively. Transcriptomics analysis can be publicly available the 30th of April and the metabolomics data have been curated. Data can soon be publicly available and the links will be provided and added to the data availability section of the manuscript.

Below in blue color, are the detailed point by point responses to the Referees' comments.

Reviewer #1 (Remarks to the Author):

In this manuscript entitled "Glycerol 3-phosphate phosphatase/PGPH-2 counters metabolic stress and promotes healthy aging via a glycogen sensing-AMPK-HLH-30-autophagy axis in *C. elegans*", the authors further determined how PGPH-2 increased longevity under glucotoxicity conditions, as a followup of their previous paper (PMID: 35017476). The authors showed that integrated pgph-2 transgenes caused physiological effects similar to those of calorie restriction, including increased lifespan, reduced fat accumulation, and improved locomotion, in addition to protecting worms from glucotoxicity. The authors then found that PGPH-2 overexpression increased healthspan and lifespan in an HLH-30-dependent manner. The authors found that glycogen reduction by pgph-2 overexpression enhanced AMPK signaling, leading to upregulation of HLH-30-mediated autophagy. They also included RNA seq and metabolomic analysis upon overexpressing pgph-2. Overall, this study is a nice followup of their recent paper in Nat. Comm. (PMID: 35017476). However, I think the information this paper add to the research field is incremental and do not believe the paper provides a conceptual leap required for Nat. Comm. Following are my specific concerns that the authors should address.

Major concerns

1) This paper has many data but the description of the data and the discussion is very loose. For example,

- a) it is not clear what is new and what is confirmation for this paper compared to the authors' previous Nat. Comm. paper.

In the previous paper, we demonstrated that the overexpression of *pgph-2/G3PP* protects from glucotoxicity and extends healthy lifespan. However, the mechanism was unknown. Importantly, in this paper we provide the mechanism by which *pgph-2 o/e* detoxifies excess glucose and reduces metabolic stress: a glycogen sensing-AMPK-HLH-30-autophagy signal transduction cascade. Only in Figure 1A-G of the manuscript, we confirm what we have previously published but now with newly-generated integrated transgenic *pgph-2* overexpressing lines (in the previously published study there was no integration of the transgene. We needed to show this and reproducing data with non-mosaic expression is important. Hence, all the rest of the figures (Figures 2-7) and supplementary figures (S1-S7) provide entirely new insight into the mechanism whereby *pgph-2 o/e* protects from excess glucose and extends healthy lifespan in the worm. Overall, we feel that the current work is not incremental as providing a mechanism for a very interesting novel enzyme related to fuel excess detoxification and aging is a very significant advance.

Below we highlight sentences in the text that were written in the previous version that help highlight the difference between what has been previously done and what is new and we also indicate the modifications that were added to the text for further clarification.

Page 4 previous text: 'However, the mechanism whereby the activation of the glycerol shunt via overexpression of PGPH-2/G3PP promotes healthy aging and protects from glucotoxicity in *C. elegans* remains to be determined.'

Page 4 previous text: 'In the present study, using *C. elegans*, we shed light on the biochemical basis of a longevity CR mimetic and healthy lifespan pathway mediated by the glycerol shunt, particularly under conditions of glucotoxicity' has been modified to: In the present study, using *C. elegans*, we shed the light on the mechanism whereby the overexpression of PGPH-2 extends healthy lifespan, particularly under conditions of glucotoxicity.

Page 6 new text. "Following the characterization of the newly-generated transgenic lines, we aimed to identify the mechanism whereby the overexpression of *pgph-2* protects from glucotoxicity and extend healthy lifespan".

Previous text in the discussion, first paragraph, "Here, we unveil the mechanism by which the activation of PGPH-2/G3PP in *C. elegans* extends healthpsan and lifespan primarily under glucotoxic conditions."

We hope that these explanations and additions address this concern of the reviewer.

- b) In addition, previous papers that are very relevant to this paper's data were not cited or discussed. For example, their data regarding glycogen should be discussed with respect to a very relevant paper (PMID: 28627510). Their data about DHAP level should be discussed with another very relevant previous paper (PMID: 26637528). Their data about AMPK should be discussed with one of the pioneering papers regarding the glucose on *C. elegans* lifespan (PMID: 17908557). These are just some examples that I easily found and many important papers are missing in the references as well. The fact that none of these key papers were not even cited is very disappointing. I strongly suggest that the authors thoroughly do literature search and revise the manuscript based on that.

We thank the reviewer for this important comment. We actively worked on quoting references that should have been included and on improving the discussion accordingly. Eighteen new references have been added to the manuscript, including all mentioned by the Referee. The discussion of our results with regards to previously published report on AMPK, glycogen, autophagy, lipophagy, calorie restriction mimetics has been much improved.

- 2)** The RNA seq and metabolomic data are not tied together with other parts and should be analyzed again as the parts of one paper.

We thank the reviewer for his concern. However, we respectfully disagree on this point but in view of this comment we now explain better the rationale of the metabolomics and RNA-seq experiments (now on Page 8 and 14). The results of the metabolomics do not show major changes in most metabolites and therefore the gene expression levels of the enzymes that regulate synthesis or degradation of these metabolites is not affected. The metabolomics data show that the overexpression of *PGPH-2* decreases Gro3P levels, particularly under glucose excess conditions. Both pieces of data (transcriptomics and metabolomics) have been performed to address separate questions. The metabolomics was performed to determine whether the activation of the glycerol shunt might lead to a severe drop in the levels of amino acids, and TCA cycle intermediates in analogy of a severe effect of calorie restriction. However we found it did not. It also aimed to determine whether the AMP:ATP and AMP:ADP ratios are increased and whether they could justify the activation of AMPK. We found it did not and therefore this drove us to the glycogen mechanism of AMPK activation. In other words, the metabolomics results successfully addressed these questions and led to two main conclusions: 1) no major deleterious effects on metabolite levels due to chronic activation of the glycerol shunt as a calorie restriction mimetic. 2) The AMP:ATP and AMP:ADP ratios are not increased and do not justify the activation of AMPK, a reason why we looked for other mechanisms of AMPK activation (glycogen-AMPK). The transcriptomics data was performed for a separate and distinct reason: determine the differential gene expression signature that could explain the protection from glucotoxicity phenotypes mediated by *pgph-2* o/e. The transcriptomics experiment was also designed to address which *pgph-2* o/e-mediated differentially expressed genes are regulated by HLH-30. We found that RNA-seq data entirely fit with our HLH-30 and autophagy data. With the current textual modifications to the results and

discussion sections, following precious comments from the three reviewers, we believe that we have made this point quite clear in this revised version.

Minor concerns

3) Please show appropriate images of Oil red O staining or fluorescent signal that match the quantification data.

We thank the reviewer for his comment. We have modified the figures of control, *pgph-2* o/e at normal and glucose excess to match better the quantification data as per the reviewers' concern. Moreover, we wish to highlight that all the quantifications are shown as dot plots for transparency. The readers can see the variations we had in all the experiments.

4) Please consider changing figure 6 to supplementary figures.

We prefer to keep figure 6 as a main figure for the following reason. As previously explained in point 2, we think that the metabolomics data are a key and integral part of the paper. The goal of this experiment was first to assess the changes in Gro3P levels and second to determine the metabolic changes that occur via the activation of the PGPH-2/G3PP, a key enzyme at the heart of metabolism, with little comprehension of its physiological roles, as it has been recently discovered (Mugabo et al., 2016, PNAS, Possik et al., 2022 Nature Communications, Al-Mass et al., 2022 Molecular Metabolism). This experiment is central to the understanding of the role of this enzyme in organismal metabolism. The results that we obtained do not show significant changes in most metabolites suggesting that the animals are coping to sustain the constant drainage in glycogen and fat reserves and are able to generate ATP. However, we acknowledge that the aim of performing this experiment was presumably not clearly stated in the results section and as previously highlighted by the reviewer in comment 2, are not clearly tied to the RNA seq data and the rest of the paper. Please refer to the answer to concern 2 where we also indicate which modifications we brought to the manuscript (pages 8 and 14) to highlight better the aim of the metabolomics data and its link with the rest of the paper.

3) Ultimately, *pygl-1* RNAi treatment is an indirect way to glycerol-generating pathway. I recommend that the authors should use *pgph-2* mutants to strengthen their results.

PYGL-1 is a glycogen phosphorylase which degrades glycogen (and not glycerol). The RNAi for *pygl-1* was used to block glycogen degradation which is not related to glycerol and determine whether the constant glycogen depletion triggers the AMPK-HLH-30 and autophagy cascade. To address this point of the referee, we now explained better the rationale about the glycogen-AMPK relationship and modified the introductory paragraph to this experiment as follow on page 15 : "AMPK is an energy sensor that activates catabolic pathways upon decreases in energy levels. Since the [AMP:ATP] as well as [ADP:ATP] ratios are not increased in *pgph-2* o/e animals in comparison to controls both under normal and high glucose conditions, we asked how is AMPK activated

independently of a change in energy drop? Recent work has shown that the AMPK-mediated energy sensing mechanism is not restricted to a drop in energy levels. Particularly, the regulatory beta subunit of AMPK has a glycogen binding domain. A growing body of research revealed an interchangeable physical and functional interaction between glycogen availability and the AMPK-dependent regulation of whole-body metabolism. Specifically, high glycogen content inhibits AMPK activity and glycogen depletion activates AMPK”.

The experiment suggested by the reviewer, which is not related to the aim of the *pygl-1* RNAi used in this paper, has in fact been already performed and the results have been presented in the previous Possik et al., Nature Communications paper published in 2022. Indeed, we demonstrated that the mutation of *pgph* enzymes severely decreases glycerol production, particularly under glycerol producing conditions such as hyperosmotic stress and glucose excess. These results are now better explained in paragraph 2 in the introduction of the current paper, when we described what we know about the PGPH enzymes from previous work: ‘In *C. elegans*, we recently identified three homologues of G3PP (PGPH-1,2,3) and have found that their protein products act as G3PP enzymes and are required for glycerol synthesis and the protection from various stresses’.

5) In figure 1, it may be more effective to combine panels D and F to show the improved locomotion of *pgph-2* o/e worms.

We thank the reviewer for his thoughtful suggestion. We switched panels 1e and 1d from the previous version and therefore put closer in proximity the locomotion experiments performed with and without glucose. However, we did not combine panels 1d and 1e together because the two experiments and their independent repeats were not conducted simultaneously. For more clarity and transparency about the data of these panels, we kept them separate, provided the raw data in data source, and put them in proximity in the figure such that the readers can compare as per the reviewers’ concern. The legends and data source have been also modified accordingly.

6) In figure 1H, it will be better to show lifespan assay data comparing *eat-2* worms and *pgph-2* o/e; *eat-2* worms as the authors did in figure 1J.

While we totally agree that this would be a great experiment to do, technically this experiment is very challenging and not possible to perform. Thus, the mutant *eat-2* animals are calorie restricted and have a strong depletion in their fat and glycogen stores. The overexpression of *pgph-2* in *eat-2* mutant animals (*pgph-2* o/e; *eat-2*) led to small, transparent animals and a strong near-sterility phenotype (Figure 1H) and matricide effects (premature death) further supporting that the depletion in energy reserves is increased when the two pathways are combined. The high number of animals required to perform glucotoxicity / lifespan curves renders the proposed experiment very challenging and technically impossible to perform as there would be not enough animals to carry on the work.

7) In figure 1H, does *pgph2 o/e* have effects on brood size under dietary restriction conditions?

It is a valid and important point and we will certainly look into it in the future. We totally agree that it is worth exploring whether the healthy longevity benefits mediated by *pgph-2 o/e* are distinct or not from previously described dietary restriction regimens. However, while this question is of high interest, we acknowledge that it will not add a substantial knowledge to the main message of the paper. Because this question could open the door to bright projects in our future work, we prefer not to tackle it rapidly in this manuscript, but rather in-depth with more thoughtfully designed experiments for follow-up studies. This is a nice project for future work and should be done with a comprehensive set of experiments in a separate paper.

8) In figure 2B, please add images for high-glucose conditions.

To address this concern, we performed a new experiment where we treated control, *pgph-2 o/e* animals, *pgph-2 o/e; hlh-30* and *hlh-30* mutant animals with 2% glucose in comparison to the control on NGM. We provided images for high glucose conditions as well as quantifications of three independent repeats. These results are now shown in Figure 2C and highlighted in the manuscript text in the result section. Figure legends and table data source have been updated accordingly.

9) Do mutants with decreased fat levels display similar effects on lifespan or autophagy like *pgph2 o/e*?

This is indeed an important question which we missed to address properly in the discussion. We now added the following paragraph to the discussion section (page17): “Autophagy, and particularly lipophagy has been clearly shown to play an important role in fat degradation. For instance, *eat-2* mutant animals (Gelino S, Plos genetics 2016; Heestand et al., Plos genetics 2013), pharmacological or genetic AAK-2 (AMPK) activation (A P Gómez-Escribano et al., pharmacological research 2020, Kosztelnik M et al., FASEB 2019, Mair et al., Nature, 2011, Moreno-Arriola et al., Plos one, 2016, Narbonne et al., Nature 2009), TOR RNAi (Blackwell et al., Genetics 2019, Lapierre et al., Nature Communications 2013), dietary restriction conditions display increased autophagy, decreased fat levels, and an extension of lifespan. However, it is important to note that while this sequence of processes has been observed in *C.elegans*, it did not always prove to be true suggesting that there are additional regulatory events that link fat catabolism to autophagy and lifespan. For instance, *daf-2* mutant animals (Insulin-signaling-like receptor) display higher fat content than wild-type, and also a heightened autophagy and extended lifespan”.

10) In figure 3A, please use a higher-resolution image for the scheme.

We made sure that the resolution of the uploaded figure is high at the process of paper revision. The problem with the previously attached version is the high compression of the full article to a reduced size PDF format, which distorted this figure. We are uploading

individual adobe illustrator figures (eps format) at this stage of the publication process to ascertain that the quality of all figures, including Fig.3A is at high/optimal resolution.

11) In figure 3B, it will be better to change labeling with number to 'upregulated' and 'downregulated'.

We added upregulated, downregulated, and unchanged to figures 3B and also did the same modification for figure S3A for consistency.

12) Please double check positioning of panel legends and consistency of legend styles.

As a final checkup of this paper, three independent readers double checked the positioning of all panel legends and consistency of legend styles to match the Nature Communications formatting guidelines.

Reviewer #2 (Remarks to the Author):

This manuscript showed PGPH-2 overexpression decreases fat accumulation, increases healthspan measured by pharyngeal pumping and locomotion, and extends lifespan under high glucose. The authors further identified TFEB/HLH-30 transcription factor activation, autophagy, AMPK activation, and glycogen storage reduction are required for PGPH-2 to benefit health. They proposed a pathway of PGPH-2—glycogen storage—AMPK—HLH-30—autophagy for the health benefits they examined in this study. The authors also analyzed gene expression and metabolites changed by PGPH-2 overexpression via RNA-seq and metabolomics. The findings are novel and of interest to the fields of aging. Most of the conclusions were well supported by experiments. I have several comments:

13) A primary concern is the physiological relevance of PGPH-2 overexpression. The study throughout this manuscript was performed using PGPH-2 overexpression transgenic strain. And they propose PGPH-2 overexpression as a CR mimetic. It would strengthen the manuscript by showing the physiological significance of increased level of PGPH-2, if the authors test whether CR induces PGPH-2 levels (RT-PCR, fluorescent reporter, or westernblot). Similar, whether AMPK activator induces PGPH-2 levels.

We thank the reviewer for his comment. Indeed, these are very interesting points and experiments. Our RNA seq data shows that the overexpression of PGPH-2 increases *eat-2* mRNA levels by 2 fold in conditions of excess glucose, now shown in Figure S3f supporting that the activation of PGPH-2 leads to an activation of a CR state, at least in excess glucose. The legends, data source and result text and figure have been updated accordingly. Furthermore, we explored previously published RNA seq showing differential expressions between *eat-2* and Wild-type and AAK-2 o/e and Wild-type (Mair et al., 2011). Interestingly, both *eat-2* mutation (as a DR model) and activation of AAK-2 are

shown to decrease mRNA levels of the *pgph-2* gene which may be observed because of the need to limit catabolism and balance the decreased levels of glycogen and fat in these animals, as all these conditions, DR, AMPK activation and PGPH-2 o/e are known to deplete energy stores. These information are now added to discussion on page 18 as follow:

“What is the physiological relevance of increased levels of *pgph2* as it may relate to AMPK activity and CR? Noticeably, we show that *pgph-2 o/e* increases *eat-2* mRNA levels by 2 fold particularly in excess glucose which is in accordance with increase pharyngeal pumping observed in *pgph-2 o/e* animals under glucose conditions. Data from previously published transcriptomics data^{63,61} showed that both *eat-2* mutation (as a CR model) and activation of AAK-2 in fact downregulate *pgph-2* gene expression levels. This is physiologically relevant as *pgph2 o/e* and the glycerol shunt act as a glucose excess detoxification machine and upon CR or enhanced AMPK activation this pathway should be shut off.”

Finally, using *pgph-2p::GFP* fluorescent reporter used in our previous paper (Possik et al., 2022, Nature Comm), we noticed that *pgph-2* is particularly induced in the dauer stage, a diapause of prolonged starvation, further supporting that PGPH-2 has physiological role in the regulation of calorie restriction and starvation, but this will be the scope of a different story.

14) PGPH-2 OE is stated as a CR mimetic in the manuscript, but it can't extend lifespan under normal glucose condition like CR. The statement of PGPH-2 OE as a CR mimetic needs to be amended.

We fully agree with the reviewer on this point. Indeed, in the original manuscript text, we tried to be cautious about this point and wrote that *pgph-2 o/e* “mimics *in part* the beneficial effects of calorie restriction”. To address the reviewers concern, we modified all the sentences related to the role of PGPH-2 o/e as a CR mimetic and stated that *pgph-2 o/e* mimics in part the beneficial effects of CR, particularly in conditions of glucotoxicity. The manuscript text has been modified in the abstract, page 4, page 6, and page 16 of the manuscript. Moreover, below in point 15, we explain our thoughts about why the overexpression of PGPH-2 extends lifespan in glucotoxic conditions and not in normal growth conditions in contrast to what has been shown in DR models including *eat-2*.

15) I would suggest more experiments to describe the healthspan and lifespan discrepancies under normal and glucotoxicity condition.

We thank the reviewer for his valuable comment. In his points listed below (a, b, c, d and e) several experiments have been performed to complete all the phenotypes with HLH-30, autophagy, and glycogen sensing, both under normal and excess glucose conditions. Overall, we believe that the level of expression of *pgph-2* and in which cell type influences the outcome on lifespan. Indeed, we observed that the overexpression of *pgph-2* using three extrachomosomal array lines extends lifespan by 7% as published in Possik et al., 2022. However, excessive or severe calorie restriction could be detrimental to the

animal's healthspan and lifespan and there is likely a threshold of calorie restriction 'without malnutrition' that promotes healthy aging. In fact, The CR mimetic action by *pgph-2 o/e* (chronic depletion of glycogen and fat stores despite normal food intake) occurs both under normal and excess glucose conditions, but since *pgph-2 o/e* animals have more energy stores fat (Fig 1 and glycogen Fig. 7) under glucose conditions in comparison to normal growth conditions, the threshold of the CR mimetic effect by *pgph-2 o/e* may enable a positive outcome on lifespan in glucotoxic and not normoxic conditions. Another possible explanation is the difference in the energy spent on reproduction, but this is not the scope of this study. Mutant *eat-2* animals have a smaller brood size in comparison to *pgph-2 o/e* and this may be also a reason why the animals live healthier and longer on normal growth conditions. Reproduction has been associated with reduced lifespan and healthspan. To clarify this point, we added the following paragraph to discussion on page 20:

"The data presented in this study differ slightly from what we have previously reported using extrachromosomal array transgenes to overexpress PGPH-2⁸. While we have previously observed a very small but significant increase in lifespan under normal conditions and no reduction in brood size, our data here show that lifespan under normal conditions is unchanged and brood size is minimally but significantly decreased by PGPH-2 *o/e*. A stable and stronger expression of *pgph-2* in the integrated transgenic strains⁷ likely explains the discrepancy in these phenotypes. Thus, with higher *pgph2* expression and glycerol shunt activation, it is anticipated that the animals will have a stronger depletion of energy stores, under normal conditions in comparison to excess glucose conditions. Particularly, fat and glycogen depots, which are critical for survival and reproduction, are strongly depleted in normal conditions and less reduced in glucotoxic conditions, which may explain the dichotomy in the outcome of *pgph-2 o/e* on lifespan. Also, the transcriptomics data hint to differences in *pgph-2o/e*-mediated gene expression regulation between normal and glucose excess conditions. RNA-seq volcano plots show a high number of differentially regulated genes in glucotoxic and not in normoxic conditions and the heat map representations reveal that ROS detoxification and unfolded protein response pathways, for instance, are only differentially regulated by PGPH-2 *o/e* in glucotoxic and not in normoxic conditions".

- a) For example, is PGPH-2-glycogen storage-AMPK-HLH-30-autophagy pathway only activated under glucose excess condition? Does *hlh-30*, *aak2*, *atg-18* mutant, or *pygl-1* RNAi blunt locomotion in normal condition (without 2% glucose)?

We have performed many experiments to address the discrepancy between healthy aging and protection from glucotoxicity (please see details below). The results of these experiments are now in main figures, and Supplementary figures. Also, textually, we brought substantial modifications to the results and discussion section. We believe that we have addressed cautiously this point and that this particular experiment is no longer necessary (please see below).

- b) The involvement of glycogen storage by *pygl-1* RNAi is examined under normal condition (without 2% glucose) for HLH-30 nuclear translocation and autophagy. Does *pygl-1* RNAi also abolish HLH-30 and autophagy activation under 2% glucose?

As per the reviewer's concern, we performed these two experiments, and indeed we show that under glucose conditions, *pygl-1* RNAi also abolishes the increased HLH-30 translocation mediated by *pgph-2 o/e*. We also show that RNAi against *pygl-1* suppresses the increased autophagic activity in *pgph-2 o/e* animals. These experiments are now shown in figures 7E and 7F. This shows that the glycogen-AMPK-TFEB-autophagy pathway is activated both under normal and glucose excess conditions. However, we agree with the reviewer that the outcome on lifespan is different and we incorporated a section in discussion to address this concern. Please see also response to point 15, section before a, b, c, d, and e.

- c) Why RNA-seq pathways are similar between PGPH-2 OE with or without 2% glucose, however the lifespan outcomes are different? Why autophagy and *hlh-30* nuclear translocation are activated in an AMPK dependent way with or without glucose, however the lifespan outcomes are different?

First we would like to mention that although the RNA-seq results may show that the two pathways are similar, this does not appear to be entirely true and we thank the reviewer for pointing this ambiguity out. We added a sentence in the discussion text to emphasize that although many of the genes are similarly differentially regulated in *pgph-2 o/e* animals in comparison to WT regardless of whether the animals are on normoxic or glucotoxic conditions, there are also subset of genes that appear to be regulated differently in normal vs excess glucose conditions. However, this is a large analysis that we will be doing in the future that is not currently the scope of this work which focuses more on the AMPK-HLH-30-autophagy pathway. Why the outcomes on lifespan are different is discussed in point 15.

We added the following paragraph in discussion on page 20: "Also, the transcriptomics data hint to differences in *pgph-2o/e*-mediated gene expression regulation between normal and glucose excess conditions. RNA-seq volcano plots show a high number of differentially regulated genes in glucotoxic and not in normoxic conditions and the heat map representations reveal that ROS detoxification and unfolded protein response pathways, for instance, are only differentially regulated by PGPH-2 *o/e* in glucotoxic and not in normoxic conditions."

- d) Is PGPH-2 only beneficial under glucotoxicity but not normal condition? These questions need to be addressed experimentally or textually.

Our data demonstrates that the glycogen-AMPK-TFEB-autophagy pathway is activated both under normal and glucose excess conditions. We demonstrate a

decreased fat and glycogen accumulation, increased HLH-30 translocation, dependence on AMPK and increased autophagic activity both under normoxic and glucotoxic conditions. However, we agree with the reviewer that the outcome on lifespan is different. Please see answer to point 15 and the modifications brought to the discussion section.

- e) Authors' thoughts on whether PGPH-2 is a beneficial gene like the statement of "CR mimetic" or PGPH-2 is only important under glucotoxicity should be discussed.

This has been addressed in details in point 15 and sub-points 15a, b, c, and d. Additional experiments have been performed and multiple paragraphs have been added to the discussion section to address this question.

16) Fig. 2C, *hlh-30* seems to be significantly lower in body bends compared to WT, how do you know it is just sick, and PGPH-2 can't reverse it? Similarly, in Fig. 4D and E, same question in terms of *atg-18*, *atg-18* KO has lower motility and survival rates in lifespans. I think proper controls need to be shown when a manipulation affects the phenotype already when compared to control. This needs to be addressed experimentally or textually.

We thank the reviewer for his concern. We performed the control experiments and have shown in supplementary figures at an earlier time point (day 4), where the animals with the mutant phenotypes are in a healthier state, that the locomotion benefits mediated by *pgph-2 o/e* are still not observed in mutants of *hlh-30* (Fig. S3a), *atg-18* (Fig. S4c), and *pygl-1 RNAi* (Fig. S7b). These mutations or downregulation shorten the lifespans in comparison to control demonstrating that the deterioration has already started to be observed. However, we would like to note that at this time-point, the differences in locomotion between control and *pgph-2 o/e* animals are small (without mutation that shortens lifespan) and this is expected, because the functional deterioration due to glucose excess and age is still at a very early stage. The manuscript text, figure legends, and data source have been updated accordingly.

We would like also to note that *daf-16* mutation also shortened lifespan (Fig. 1), and the advantage mediated by *pgph-2 o/e* at the 9th and 12th day time-point was still observed and this is why we concluded that the healthy longevity pathway is independent from insulin signaling. In these epistasis experiments, the majority of the population from the mutant groups is still alive at day 12 and the median for death events in approximately around day 15, which is 3 to 5 days after the number of body bends per second have been recorded. Our data strongly support that *hlh-30*, *aak-2*, *atg-18*, *pygl-1* are required for the increased healthy aging mediated by *pgph-2 o/e* under excess glucose.

17) Fig. 5C westernblot result is not solid. AMPK protein level needs to be shown. Quantification and statistics need to be shown based on p-AMPK/AMPK ratio.

The available AMPK antibodies do not recognize AMPK in *C.elegans* due to the distance in the protein conformation protein across species mouse/rats/humans and worms. Only

the p-AMPK antibody recognizes the catalytic Thr172 site, where we show the increased p-AMPK levels in *pgph-2 o/e* animals in comparison to the controls. Therefore, the protein levels of AMPK cannot be measured to calculate the p-AMPK/AMPK ratio. We have encountered the same comment by other referees in three previous publications. Similarly to what we and others have published in the field to address this type of concern, we provided mRNA levels of the two AMPK catalytic subunits AAK-1 and AAK-2 in Figure S6b and showed that the AMPK mRNA levels are indeed unchanged (For reference, please see publications by Scultz et al., 2007 Cell metab, Mair et al., 2015, Cell, Possik et al., 2014, Plos genetics). We also provided a dotted bar graph plot showing the quantifications from four independent experiments to support our result on AMPK activation (Fig. 5c). Overexpression of *pgph-2* leads to a significant activation of AMPK on phospho-Thr172 (two AAK-2 isoforms; Jin-Hyuck Jeong, Nature Communications 2023) detected with four independent repeats (p=0.026). Gel pictures for the four independent repeats have been also provided as requested by the journal for increased transparency. We would like to note that starvation of the control strain was used as a positive control for pAMPK induction. While this induction has been repetitively observed in the independent repeats, the level of induction varied between experiments and led to a non-significant p-value (p=0.07). However, the effect of starvation on pAMPK activation in all model organisms has been widely shown and described. The manuscript legends have been modified accordingly. In summary, because of the above considerations and the AMPK genetics data that we have included in the paper, we think that the AMPK data are conclusive.

18) Authors showed RT-PCR results of *lgg-1*, *lgg-2*, *atg-18* under PGPH-2 OE in Fig. S4A. Any autophagy genes including these three were revealed in the RNA-seq data? This needs to be described and discussed.

The results for the autophagy genes that are present in Figure S4 are obtained from RNA-Seq. We clarified this issue in the legends. Additionally, we added supplementary table 10 where we show the fold difference for all the listed autophagy genes that could be detected with RNA-Seq (*atg-2*, *atg-3*, *atg-4.1*, *atg-4.2*, *atg-5*, *atg-6*, *atg-7*, *atg-8.1* or *lgg-1*, *atg-8.2* or *lgg-2*, *atg-9*, *atg-10*, *atg-13*, *atg-16.1*, *atg-16.2*, and *atg-18*). The following genes *atg-4.1*, *lgg-1*, *lgg-2* and *atg-18* were significantly upregulated in *pgph-2 o/e* animals in comparison to WT and required *hlh-30* and are shown in Figure S4 as well. The manuscript result text and legends have been updated accordingly to the Referee comment. Not observing a bigger panel of autophagy genes upregulated in *pgph-2 o/e* animals in comparison to WT could be due to the animal stage (L4/young adult animals for RNA-Seq) and one day adult for autophagy assays. Also, other reasons could include the fact that the RNA-Seq was done with whole worm extracts which may dilute the signal if the upregulation of the autophagy genes occurs in certain tissues.

Minor points:

19) Fig. 1A,B,C, Fig. 5A, Fig. S1A: please use dot scared plots.

As requested by the reviewer and the nature editorial board, we modified Figures 1A, B, C, 5A and S1A and used dot scared representations.

20) Fig. 1D, F, H: please use a different color for error bars of control and PGPH-2 OE, it confuses people by use the reciprocal color of the comparing condition.

According to the reviewers' comment, we used a blue color to represent the error bars in Figures 1D, F, and H. We note that following the comment of reviewer 1, figure 1F is now labeled as figure 1D.

21) Page 6, “with a marginal effect on lifespan (Figure 1D-E)”. please add “non-significant” or describe it as no effect.

We thank the reviewer for this comment. We modified accordingly ‘with a marginal effect on lifespan’ to ‘non-significant’ effect on lifespan.

22) Page 6, “and also medium and maximal lifespan under glucotoxicity”. I believe “increase” is missing in the sentence.

We thank the reviewer for this observation. Indeed, the word increase was missing. We corrected accordingly: and also increases medium and maximal lifespan under glucotoxicity.

23) Page 6, “In addition, the newly generated transgenic line phenocopies what we have observed using extrachromosomal arrays [7]”. Please rephrase this to be clearer to a non-worm reader so they know what the difference is here between the transgenic line and extrachromosomal arrays generated lines.

We thank the reviewer for his thoughtful comment. To avoid any confusion we added the following sentence at the beginning of this paragraph: “In our previous work, we generated transgenic lines overexpressing *pgph-2* and empty vector (EV) via DNA injection with co-injection marker plasmids expressing mCherry in the pharynx to form multi-copy extrachromosomal arrays, which get transmitted to the next generation via partial and variable transmission rates and mosaic expression in different cell types. In the current work, we integrated these transgenes to the worm genome to form stable lines overexpressing *pgph-2* or EV used as control and outcrossed them more than seven times to the wild-type (WT) strain and observed that *pgph-2* gene mRNA levels are induced by about six folds (Figure S1A)”. In addition, we rephrased the sentence highlighted by the reviewer to: “In addition, the newly generated *pgph-2* overexpressing stable transgenic line phenocopies what we have previously observed using the *pgph-2* overexpressing extrachromosomal array transgenes”.

24) Page 6, “Noticeably, the brood size decrease by PGPH-2 o/e is minimal in comparison to the severe brood size decrease observed in *eat-2* animals in normal growth and excess glucose conditions (Figure S1D)”. What does this sentence mean in line with the conclusion of this panel? The use of “minimal” is confusing.

We meant by this sentence that the brood size of *eat-2* is much smaller than that of *pgph-2* o/e animals. To avoid any confusion and according to the reviewers comment, we rephrased this sentence to the following: “Noticeably, the overexpression of *pgph-2* led to a smaller decrease in brood size in comparison to the severe brood size decrease observed in *eat-2* animals in normal growth and excess glucose conditions (Figure S1D)”.

25) Page 7, “not due to a transgene artefact”, artefact is a typo.

We modified ‘artefact’ to ‘artifact’.

26) Page 11, “preventing the transcription of the CLEAR network,” Please spell out CLEAR.

We spelled out CLEAR on the first time we mentioned it in the text as CLEAR (Coordinated Lysosomal Expression and Regulation).

27) Fig. S5B and C, needs quantification and statistics.

We added quantification and statistics to Fig. S5. Figure legend and data source have been modified accordingly.

28) Fig. 5B, color legend in the figure is missing.

We thank the reviewer for this observation. We added the color legend to Fig. 5B.

Reviewer #3 (Remarks to the Author):

This manuscript by Possik *et al.* reports critical insights into the role of G3PP/PGPH-2 in high glucose diet in *Caenorhabditis elegans*. The authors demonstrate that the overexpression of *pgph-2* activates a cascade of events on the AMPK-TFEB-autophagy axis that might serve as an alternative to healthy ageing by overcoming the side effects of calorie restriction, such as reduced fertility. The manuscript presents answers to scientifically relevant hypotheses supported by clear, compelling figures. However, some sections should be reconsidered to alleviate minor concerns—outlined below—which could be resolved by reanalysing parts of the data.

Minor concerns

29a) The methods section for bioinformatic analysis states that the authors used the *featureCounts* function (from *Rsubread* package) to obtain gene counts. It is not clear whether the raw data was mapped or aligned to the reference genome before feature

counting or skipped alignment using tools such as *salmon* that is bias aware (with `--gcBias` parameter). The reference genome version should also be added (i.e., GenBank or RefSeq assembly accession) for either method.

29b) The authors used *DESeq2* package for conditions/treatments with biological replicates and *edgeR* package for conditions without biological samples to analyse RNA-seq data. Although these choices were appropriate analysis methods, the type of input data (RPKM) is not appropriate for the *DESeq2* method (See original paper PMID: 25516281). First, RPKM should be used with single-end reads, while FPKM is used for paired-end RNA-seq data (though the legend for Figure S2A states FPKM was used). Given the authors analysed paired-end data, the usage of FPKM would be more accurate (See PMIDs: 32284352; 34158060). Second, despite their popularity, these units (i.e., RPKM, FPKM and TPM) should rather be used for qualitative assessments such as Principal Component Analysis (PCA) but not for differential expression analysis.

For comments 29a and b, we acknowledge the reviewer's thoroughness in the revision. This is a very important point. The entire method for RNA-seq has been carefully revised. The method written in the initial version of the paper, did not fully fit the final report that we received from the Novogene company. This error stemmed from a miscommunication problem with the Novogene team that sent us a standard protocol for us to write the methods that is inconsistent with the final report. We attached the final study report for transparency about the RNA-seq data processing as a reviewer-only document. We also note that the report as well as all raw and final data are now available in GEO, a publicly accessible database.

29 a) According to the reviewer's comment, we revised the methods RNA-seq and the report is attached as well in the data repository (GEO). We added a section in methods called RNA-seq data analysis in which we explained how the raw reads were mapped and aligned to the reference genome before feature count analysis. This section is copied below:

"Reference genome and gene model annotation files were downloaded from genome website directly. Index of the reference genome was built using Hisat2 v2.0.5 and paired-end clean reads were aligned to the reference genome using Hisat2 v2.0.5. We selected Hisat2 as the mapping tool for that Hisat2 can generate a database of splice junctions based on the gene model annotation file and thus a better mapping result than other non-splice mapping tools. The mapped reads of each sample were assembled by StringTie (v1.3.3b) in a reference-based approach. Feature Counts v1.5.0-p3 was used to count the reads numbers mapped to each gene. And then FPKM of each gene was calculated based on the length of the gene and reads count mapped to this gene. FPKM, expected number of Fragments Per Kilobase of transcript sequence per Millions base pairs sequenced, considers the effect of sequencing depth and gene length for the reads count at the same time, and is currently the most commonly used method for estimating gene expression levels. Differential expression analysis of two conditions/groups (two biological replicates per condition) was performed using the DESeq2 R package (1.20.0). DESeq2 provide statistical routines for determining differential expression in digital gene expression data using a model based on the negative binomial distribution. The resulting

P-values were adjusted using the Benjamini and Hochberg's approach for controlling the false discovery rate. Genes with an adjusted P-value ≤ 0.05 found by DESeq2 were assigned as differentially expressed".

29 b) The type of input data for DESeq2 method is indeed FPKM and not RPKM as the Seq is a paired-end data. We did not use RPKM. This has been clarified in the methods section.

30) The authors analysed four biological replicates per condition for RNA-seq experiments (Figure S2)—though the methods state 'three' replicates. The reproducibility problem in *C. elegans* research has been addressed in several reviews, including Urban *et al.* (2021) [PMID: 34793939] and Pho & MacNeil (2019) [PMID: 31127069]. For instance, two out of four replicates in the *hlh-30* + *Glucose* condition have differential R^2 values than the other two replicates (Figure S2B). Therefore, it would be more convincing and conclusive if the authors computed power analysis to show the number of replicates per condition was robust enough.

We thank the reviewer for noticing this mistake. As shown in figures and PCA plots, the number of biological replicates is four. We corrected the mistake in methods and indicated four biological replicates instead of three. As per the robustness of the method, we discussed with *C.elegans* and mammalian transcriptomics experts and we consulted a biostatistician to compute power analysis as per the reviewer's concern. R package 'pwr' (Basic Functions for Power Analysis, version 1.3-0) function 'pwr.anova.test' was used for assessing the power and robustness of the RNA-seq experimental design. With 32 samples, 4 replicates per condition for 8 conditions, assuming a medium effect size of 0.25, there is 83.7% power for rejecting the null hypothesis when in fact it is false (avoid Type II error). This percentage falls within the acceptable norm (80-90%) (Banerjee *et al.*, 2009; Ceyhan Ceran Serdar *et al.*, 2021).

Moreover, following a series of discussions, we came to the following conclusions:
1- It is true that reproducibility with *C.elegans* experiments is sometimes arguable, in general, it is a model with more robust reproducibility in comparison to other models like mice for instance. In general, when performing omics experiments on samples biologically pooled from the same experiment, the variability with *C.elegans* is low in comparison to samples from rodent tissues where variability is higher.

2- To balance data quality and cost, in *C.elegans*, the field uses in general four replicates per sample for RNA seq. We list below a few papers published in high impact journals that used the *C.elegans* model for transcriptomics analysis, and performed their analyses also with four biological replicates per sample (Viviskis *et al.*, 2017, Immunity, Khursheed A Wani, *et al.*, 2021, Elife, Kumar *et al.*, 2022 Science advances, El Houjeiri *et al.*, 2019, Cell reports, Weir *et al.*, 2017, Cell metabolism and many others).

3- The differential gene expression is accompanied with a p-value, and our interpretations are based on statistical significance $p < 0.05$ and genes that are significantly changed by 2 fold or higher. The computed power analysis will not change much the interpretation of what we have identified and discussed in the paper, which are highly relevant to explain the results obtained with genetics experiments. We agree that if sample size was higher

than 4 (6 for example), we probably would have been able to detect more pathways with higher confidence and resolution, but the data that is already present in the manuscript (when looking at p-values) and significantly changed genes and pathways, explains well enough the biology of *pgph-2 o/e*. We were careful with our claims in the manuscript and we made sure not to make conclusion on pathways or genes that have not been significantly changed.

4- At the end, to balance cost and data resolution, we believe that four samples is robust enough to make solid conclusions with affordable cost (32 samples instead of 48, if we would have chosen a scenario of 6 replicates). For transparency, and to address the concern of the reviewer, we mentioned the computed power analysis in the methods.

31) Also, a 2-dimensional PCA plot would be easier to comprehend the variation within a condition than the 3-dimensional PCA plot (Figure S2C) since the third principal component is rather low (5.45%).

We agree with the reviewer on this point, and as per this recommendation, we modified Figure S2C to show a two-dimensional PCA plot instead of a three-dimensional plot.

Other recommended refinements:

32) Reporting of statistics: The authors reported experimental results with transparency (visible individual data points) and the appropriate type of statistical tests. Findings were visualised by using average values and standard error of the means (mean \pm SEM). Although mean \pm SEM continues to be reported in life sciences research, SEM does not describe the sample. Therefore, the mean values should be reported with standard deviation (SD), not SEM (PMIDs: 33402813, 23125963). Given the robustness of the experimental procedures and results, changing SEM to SD will not affect their outcomes. Nevertheless, it would improve the quality of the manuscript from a statistics point of view.

We thank the reviewer for his positive comment and are indeed pleased that she/he noticed the high level of transparency and robustness of the work. As per this statement, changing SEM to SD will not affect the result outcome for this paper. However, for future manuscripts, we will certainly use SD instead of SEM when appropriate.

33) Adding software versions and data availability could improve the reproducibility of the analysis.

As per the reviewers comment, we indicated all the software names and versions used for RNA seq analysis in table S12. Also a sentence about the availability of this table has been added to the methods section. The RNA seq report, raw values, and final data have been also uploaded to the GEO database and will be publicly available prior to the publication, as per the Nature Portfolio request.

34) The data source for Figure 1F (the exact number of tracks) is missing.

We thank the reviewer for noticing the missed data source. We added the raw data for Figure 1F.

35) The legend of Figure 7 for some of the subplots is mislabelled/missing:
Original legend:

“Figure 7: PGPH-2 *o/e* reduces glycogen stores to activate an AMPK-dependent/HLH-30/autophagy cascade. A-B) Representative iodine staining images and quantification in control and *pgph-2 o/e* animals in normal growth conditions (A) and plates supplemented with 2% glucose (B). (C) HLH-30 nuclear translocation in WT and *pgph-2 o/e* animals expressing the HLH-30::GFP transgene exposed to Ev and *pygl-1* RNAi for two generations. Data represent mean \pm SEM from three independent experiments. (D) Representative confocal images and quantification of LGG-1::mCherry puncta in the intestines of day 1 WT and *pgph-2 o/e* exposed to Ev and *pygl-1* RNAi for two generations. Data represent mean \pm SEM from three independent experiments. The scale bar indicates 20 μ m. (E-F) Locomotion analysis on days 9 and 12 of age, measured by body bend per second in control and *pgph-2 o/e* exposed to Ev and *pygl-1* RNAi bacteria and grown on plates supplemented with 2% glucose. Data is shown using dot plots with denoted mean \pm SEM from three independent experiments. P-values are obtained by one-way ANOVA with the Bonferroni correction. E) Lifespan of control and *pgph-2 o/e* exposed to Ev and *pygl-1* RNAi bacteria and grown on plates supplemented with 2% glucose. The number of separate lifespan experiments, animals and detailed statistics are shown in Supplementary Table S1. P-value is obtained using the two-sided Mantel-Cox test.”

Suggested legend:

“Figure 7: PGPH-2 *o/e* reduces glycogen stores to activate an AMPK-dependent/HLH-30/autophagy cascade. A-B) Representative iodine staining images and quantification in control and *pgph-2 o/e* animals in normal growth conditions (A) and plates supplemented with 2% glucose (B). **C) Glycogen degradation pathway. D) Representative iodine staining images and quantification in WT and *pgph-2 o/e* exposed to Ev and *pygl-1* RNAi. E)** HLH-30 nuclear translocation in WT and *pgph-2 o/e* animals expressing the HLH-30::GFP transgene exposed to Ev and *pygl-1* RNAi for two generations. Data represent mean \pm SEM from three independent experiments. **F) Representative confocal images and quantification of LGG-1::mCherry puncta in the intestines of day 1 WT and *pgph-2 o/e* exposed to Ev and *pygl-1* RNAi for two generations. Data represent mean \pm SEM from three independent experiments. The scale bar indicates 20 μ m. **G) Locomotion analysis on days 9 and 12 of age, measured by body bends per second in control and *pgph-2 o/e* exposed to Ev and *pygl-1* RNAi bacteria and grown on plates supplemented with 2% glucose. Data is shown using dot plots with denoted mean \pm SEM from three independent experiments. P-values are obtained by one-way ANOVA with the Bonferroni correction. **H) Lifespan of control and *pgph-2 o/e* exposed to Ev and *pygl-1* RNAi bacteria and grown on plates supplemented with 2% glucose.”******

We thank the reviewer for the thoroughness in the reading of this manuscript and for noticing this mistake in legends. We revised the legend of figure 7 according to the reviewer's suggestions.

36) The manuscript addresses the role of *pgph-2* on the lifespan and healthy ageing through the enhanced activity of PGPH-2 and of the glycerol shunt which leads to constitutive activation of AMPK, HLH-30 and autophagy, mimicking only beneficial effects of calorie restriction. Findings on the unexplored role of PGPH-2 and the glycerol shunt demonstrate protection against ageing and promote fit and healthy longevity. This study by Possik *et al.* will contribute indispensable discoveries to ageing research on *C. elegans* literature.

Thank you for your positive input and insightful suggestions which certainly strengthened the manuscript.

REVIEWERS' COMMENTS

Reviewer #1 (Remarks to the Author):

The manuscript has been improved through revisions. Unfortunately however, I cannot change my initial assessment about the manuscript and this is still a nice followup of their previous paper without conceptual leap.

Reviewer #2 (Remarks to the Author):

What are the noteworthy results?

Answer: The revision is substantial, and addressed all my concerns.

Will the work be of significance to the field and related fields?

Answer: Yes

How does it compare to the established literature? If the work is not original, please provide relevant references.

Answer: It will be a valuable addition to the field.

Does the work support the conclusions and claims, or is additional evidence needed?

Answer: Yes, the work support the conclusions.

Are there any flaws in the data analysis, interpretation and conclusions? Do these prohibit publication or require revision?

Answer: The revision has a great improvement.

Is the methodology sound? Does the work meet the expected standards in your field?

Answer: Yes. Yes.

Is there enough detail provided in the methods for the work to be reproduced?

Answer: Yes. One question I have in the methods part of pumping and locomotion is the age of the worms. I suppose the day of age is day of adulthood but not from birth? e.g., is pumping rate of day 1 measured in figure 1b day 1 of adulthood or day 1 from birth as a stage 1 larvae?

The authors have addressed my comments by modifying and clarifying data analyses in the text.

Minor comment:

In the rebuttal, authors mentioned that “With 32 samples, 4 replicates per condition for 8 conditions, assuming a medium effect size of 0.25, there is 83.7% power for rejecting the null hypothesis when in fact it is false (avoid Type II error). This percentage falls within the acceptable norm (80-90%) (Banerjee et al., 2009; Ceyhan Ceran Serdar et al., 2021).” And “For transparency, and to address the concern of the reviewer, we mentioned the computed power analysis in the methods.” Maybe I missed it, but I didn’t find the power analysis mentioned in the methods. If it is not included, please include it into the methods.

Point by point response to the reviewers (NCOMMS-22-47059A)

Reviewer #1 (Remarks to the Author)

The manuscript has been improved through revisions. Unfortunately, however, I cannot change my initial assessment about the manuscript and this is still a nice followup of their previous paper without conceptual leap.

We thank the reviewer for his valuable comments. The advice certainly improved the manuscript.

Reviewer #2 (Remarks to the Author)

- What are the noteworthy results?
Answer: The revision is substantial, and addressed all my concerns.
- Will the work be of significance to the field and related fields?
Answer: Yes
- How does it compare to the established literature? If the work is not original, please provide relevant references.
Answer: It will be a valuable addition to the field.
- Does the work support the conclusions and claims, or is additional evidence needed?
Answer: Yes, the work support the conclusions.
- Are there any flaws in the data analysis, interpretation and conclusions? Do these prohibit publication or require revision?
Answer: The revision has a great improvement.
- Is the methodology sound? Does the work meet the expected standards in your field?
Answer: Yes. Yes.
- Is there enough detail provided in the methods for the work to be reproduced?
Answer: Yes. One question I have in the methods part of pumping and locomotion is the age of the worms. I suppose the day of age is day of adulthood but not from birth? e.g., is pumping rate of day 1 measured in figure 1b day 1 of adulthood or day 1 from birth as a stage 1 larvae?
- The authors have addressed my comments by modifying and clarifying data analyses in the text.

We thank the reviewer for the valuable comments that substantially improved the manuscript. Concerning the last underlined minor concern, we added the age of the worms as days from adulthood not from birth. This is now clarified in the methods section for pharyngeal pumping and locomotion.

Reviewer #3 (Remarks to the Author)

Minor comment:

In the rebuttal, authors mentioned that “With 32 samples, 4 replicates per condition for 8 conditions, assuming a medium effect size of 0.25, there is 83.7% power for rejecting the null hypothesis when in fact it is false (avoid Type II error). This percentage falls within the acceptable norm (80-90%) (Banerjee et al., 2009; Ceyhan Ceran Serdar et al., 2021).” And “For transparency, and to address the concern of the reviewer, we mentioned the computed power analysis in the methods.” Maybe I missed it, but I didn’t find the power analysis mentioned in the methods. If it is not included, please include it into the methods.

We thank the reviewer for the valuable comments and for noticing that the power analysis that we performed during the revision was not mentioned in methods. As per the reviewer’s request, we verified that it is now included in the methods section.